# Accumulation of mutations in antibody and CD8 T cell epitopes in a B cell depleted lymphoma patient with chronic SARS-CoV-2 infection

Antibodies against the spike protein of severe acute respiratory syndrome coronavirus-2 (SARS-CoV-2) can drive adaptive evolution in immunocompromised patients with chronic infection. Here we longitudinally analyze SARS-CoV-2 sequences in a B cell-depleted, lymphoma patient with chronic, ultimately fatal infection, and identify three mutations in the spike protein that dampen convalescent plasma-mediated neutralization of SARS-CoV-2. Additionally, four mutations emerge in non-spike regions encoding three CD8 T cell epitopes, including one nucleoprotein epitope affected by two mutations. Recognition of each mutant peptide by CD8 T cells from convalescent donors is reduced compared to its ancestral peptide, with additive effects resulting from double mutations. Querying public SARS-CoV-2 sequences shows that these mutations have independently emerged as homoplasies in circulating lineages. Our data thus suggest that potential impacts of CD8 T cells on SARS-CoV-2 mutations, at least in those with humoral immunodeficiency, warrant further investigation to inform on vaccine design.

Soon after SARS-CoV-2 emerged in late 2019, national and international collaborative efforts have allowed sequencing data of full-length SARS-CoV-2 genomes to be shared immediately, allowing their use for molecular epidemiological studies and for predictions of structural changes of specific viral proteins important for immune recognition. Infection of immunocompetent individuals elicits a complex virus-specific immune response, which in most cases includes antibodies and memory B cells specific for the spike protein[1] and nucleoprotein[2], and can include antibodies against ORF8 or other non-structural proteins[3,4]. Infection also gives rise to a broad response of antigen-specific CD4 and CD8 T cells directed against multiple structural and non-structural proteins[5–7]. On average, the combined CD4 and CD8 T cell memory response of a given individual encompasses about 30 epitopes[5,8]. While the spike-specific T cell response is dominated by CD4 T cells, nucleoprotein tends to elicit particularly pronounced CD8 T cell responses[6,9]. Robust SARS-CoV-2 specific CD8 T cells have been detected in acute infections and convalescents of mild and asymptomatic cases[10], but also in severe cases at high frequency and breadth[9].

Due to the complex nature of the SARS-CoV-2-specific immune response in immunocompetent individuals, the individual contributions of its constituents to protection are difficult to separate. Protection from severe disease was described to correlate with a coordinated response of antibodies, CD4 and CD8 T cells[2]. While it is accepted that neutralizing antibodies protect from infection and provide correlates of protection against severe disease[11,12], virus-exposed individuals sometimes develop specific T cells in the absence of a detectable antibody response[10,13]. An important role of CD8 T cells was further suggested by a rhesus macaque model where depletion of CD8 T cells abrogated protective immunity against rechallenge with SARS-CoV-2 in the setting of waning antibody levels[14], and by a mouse model where T cell responses elicited by vaccination were protective against severe disease[15]. In humans, it was observed that the CD8 T cell response to a single immunodominant nucleoprotein epitope is

✉ e-mail: elham.khatamzas@med.uni-heidelberg.de

associated with less severe disease, and T cells with this specificity have unusually high frequency in the naive repertoire[16,17]. These findings also suggest that individual predispositions and the precise specificity of the T cell response down to the level of individual epitopes should be considered in evaluation of the protective role of CD8 T cells. The functional importance of T cells in different stages of infection and disease continues to be intensely discussed[8,18].

Antibodies to the spike protein have shaped evolution of SARS-CoV-2 to a great degree, in particular by enforcing mutations in spike epitopes targeted by neutralizing antibodies. The first variant classified by the WHO as variant of concern (VOC) was the alpha variant with increased transmissibility and pathogenicity[19,20]. The alpha variant evaded recognition of neutralizing antibodies targeting the N-terminal domain (NTD) of the spike protein, but nonetheless showed relatively little escape from recognition by total plasma antibodies elicited by infection with ancestral SARS-CoV-2[21,22]. Subsequent variants beta, gamma, and delta developed stronger resistance to antibody-mediated recognition and neutralization due to various new mutations in the NTD and receptor-binding domain (RBD)[21–23]. The largest number of mutations across the spike protein then appeared in the much more highly transmissible omicron variants[24], contributing to these variants' massively enhanced spread even in previously infected or vaccinated individuals[25] by means of antibody evasion[26–28] and altered molecular and cellular infection pathways of omicron strains[29].

One hypothesis is that such variants may have evolved in immunocompromised patients with prolonged viral replication in chronic infections[30], although evidence corroborating this is lacking so far. In immunocompetent hosts infection is typically resolved within a few days to weeks with low intra-host diversity of SARS-CoV-2[31,32]. This contrasts with chronic infections in immunocompromised patients where prolonged viral replication promotes within-host diversification[33–35]. Multiple case reports of chronic infections have shown adaptive evolution of SARS-CoV-2 with a focus on immune escape mutations of the spike protein to neutralizing antibodies[36].

A number of different underlying immunocompromising conditions have been reported for cases with chronic SARS-CoV-2 infection, including advanced HIV infection[37], immunosuppressive therapy for solid organ transplantation[38] and autoimmune conditions[35] as well as immunological disorders due to hematological cancer and its treatment[39,40]. This indicates that different dysregulations of the immune system can result in delayed or insufficient viral clearance, highlighting the relevance of a well-coordinated humoral and cellular immune response for disease control. Interestingly, a study of SARS-CoV-2-infected cancer patients found no association between impaired B cell function and increased mortality, while B cell depleted patients with greater numbers of CD8 T cells had better survival[41]. These data suggest that CD8 T cells can contribute to successful immune control of SARS-CoV-2 as already known for other viral infections such as human immunodeficiency virus (HIV)[42] and hepatitis B and C[43]. In infections with these viruses, immunity can be jeopardized by the emergence of immune escape mutations in T cell epitopes[44–46]. The mechanisms of reduced T cell recognition include mutations in the epitope region, which is recognised by the T cell receptor or in the anchor residues of the viral peptide that binds to the MHC molecules interfering with antigen presentation, as well as other escape pathways such as mutations affecting proteasomal processing[47]. To date it remains unclear if selection pressure mediated by CD8 T cells can drive mutational escape of SARS-CoV-2.

Here we chronicle intra-host evolution of SARS-CoV-2 over more than 150 days in a patient with severe humoral immune deficiency receiving convalescent plasma therapy. We observe the emergence of mutations located in antibody and CD8 T cell epitopes that result in reduced immune recognition. These findings suggest the hypothesis that, at least in hosts with deficient humoral immunity, T cell selection pressure might contribute to adaptive evolution of SARS-CoV-2.

## Results

### Chronic SARS-CoV-2 infection in a lymphoma patient on B cell depleting immunochemotherapy

A woman in her seventies was hospitalised in spring 2020 with COVID-19 and severe acute respiratory distress syndrome (ARDS). She was managed in our intensive care unit developing multiple complications and multi-organ failure until her death five months later. Her past medical history included follicular lymphoma treated with standard chemotherapy consisting of cyclophosphamide, doxorubicin, vincristine and prednisolone in combination with the B cell-depleting anti-CD20 antibody obinutuzumab up to a month prior to presentation. As expected, standard ELISAs (Euroimmune) for IgG and IgA against the S1 domain of the SARS-CoV-2 spike protein were negative throughout the course of disease. Furthermore, both immunoglobulin levels and lymphocyte subsets measured in peripheral blood indicated severe immunosuppression (Supplementary Tables 1, 2). COVID-19-directed treatment included steroids and multiple doses of convalescent plasma (Fig. 1a). Therapy with remdesivir was deemed unsafe due to severe renal failure and known hypersensitivity to the vehicle sulfobutylether-β-cyclodextrin. Virological studies revealed persistently high SARS-CoV-2 RNA levels (Fig. 1b) with replication-competent virus in respiratory tract specimens until death 156 days following COVID-19 diagnosis.

### Intra-host evolution of SARS-CoV-2

In order to study intra-host viral evolution, sequential upper and lower respiratory tract specimens were used to generate 21 near full length genomes of SARS-CoV-2 spanning 150 days (Supplementary Table 3). The infecting strain was classified as B.1.1.29 (20B) bearing the spike mutation D614G and clustered with other sequences circulating at that time (Fig. 2a). Over the following 5 months the virus diversified extensively with 34 different mutations in coding regions emerging at >40% frequency abundance (Supplementary Fig. 1). To study viral evolution in adaptation to the host adaptive immune response, we focused our analyses on the 12 non-synonymous mutations that became fixed in the dominant variants until death (Fig. 2b).

### Antibody neutralization of mutant SARS-CoV-2 isolates

We recognised three mutations located in the spike gene previously associated with antibody escape: A deletion at position 144 in the N-terminal domain (NTD) (mutation I), and S477N (mutation III) and E484K (mutation II) substitutions in the receptor binding domain (RBD) (Fig. 3a). We performed in vitro neutralization assays testing the activity of serum samples taken from the donors of convalescent plasma against patient isolates. All but one donor showed robust neutralizing responses against an early pandemic isolate carrying only the D614G spike mutation (MUC-IMB01) and an autologous isolate sampled at day 20, before the emergence of the additional mutations in the spike gene (Fig. 3b). However, the autologous variant isolated at day 83 carrying the del144 and E484K spike mutations was only weakly neutralized by one donor serum and none of the others tested, showing that the combination of these RBD and NTD alterations was sufficient to almost completely abrogate neutralizing activity. Structural analysis illustrated that the two mutations in the RBD and the deletion in the NTD are located in epitopes targeted by exemplary neutralizing antibodies REGN10993, Ly-COV555, and 4A8 (Fig. 3c). Taken together, these data suggest that in this severely immunodeficient patient the neutralizing polyclonal antibodies administered with the convalescent plasma transfusions may have mediated immune selection pressure that drove adaptation of the viral spike protein to evade these responses.

### SARS-CoV-2-specific CD8 T cell responses

We investigated whether the lack of an effective T cell response might have contributed to the failure to clear SARS-CoV-2 infection. An IFN-γ

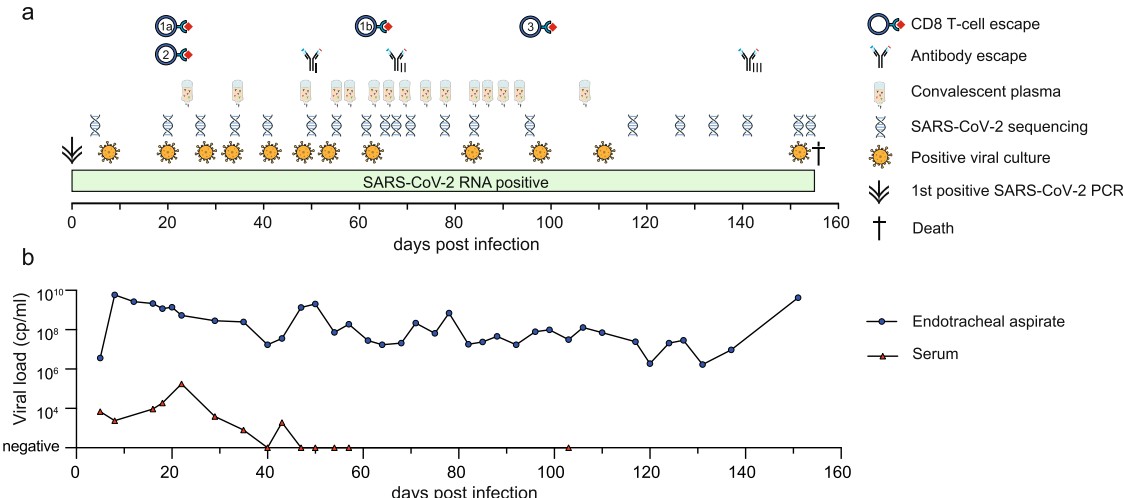

**Fig. 1 | Timeline of events and longitudinal viral load results. a** Chronology of events in relation to the time point of COVID-19 diagnosis by the first positive SARS-CoV-2 RNA PCR result (day 0). The diagram includes the time points of administration of convalescent plasma, of positive viral cultures from patient respiratory samples on Vero E6 cells, of SARS-CoV-2 genome sequencing, and of first detection of the potential CD8 T cell (1a, 1b, and 2, and 3, see also Fig. 2b) and antibody escape mutations (I, II and III) which are further investigated in this study are indicated. **b** SARS-CoV-2 RNA copy numbers per ml of endotracheal aspirates and serum are shown over time. Days are indicated in relation to the first positive SARS-CoV-2 RNA result (day 0). Source data are provided in the Source Data file.

ELISPOT assay that was performed with a fresh peripheral blood sample on day 127 showed that the patient had strong T cell responses to antigens spike (N-terminal and C-terminal half), nucleocapsid, and ORF3a (all derived from Wuhan-Hu-1 D614) (Fig. 4a). For spike and ORF3a, the proportion of IFN-γ-secreting antigen-specific T cells in peripheral blood mononuclear cells was higher in the patient compared to five immunocompetent convalescent donors at a median of 55 days post-infection (Fig. 4a). Nonetheless, analysis of peripheral blood on day 145 by flow cytometry demonstrated deficiencies in lymphocyte subsets. While numbers of monocytes were normal (347/µl), both lymphocytes (213/µl) and T cell numbers (142/µl) were reduced. T cells were predominantly CD8-positive (60%), and many CD8 T cells (31%) co-expressed the activation markers CD38 and HLA-DR, whereas CD4 T cells were less activated (Fig. 4b).

### Effect of mutations on CD8 T cell recognition of SARS-CoV-2 epitopes

Since the patient had strong T cell responses to non-mutant SARS-CoV-2 antigens and activated CD8 T cells, we hypothesised that mutations in SARS-CoV-2 might have emerged under selective pressure by CD8 T cells and had led to inactivation of T cell epitopes. Therefore, we examined whether these mutations were located in potential CD8 T cell epitopes presented by the patient's HLA class I molecules (Supplementary Table 4). For epitope prediction, we used an approach based on simple anchor motifs (Supplementary Table 5). Out of twelve non-synonymous mutations that became dominant in the patient's SARS-CoV-2 variants, nine altered the amino acid sequences of putative CD8 T cell epitopes (Supplementary Table 6), including the eight earliest mutations (day 41-97). Both nucleocapsid mutations affected the same nonameric B*35:01-restricted candidate epitope with pre-mutation sequence TPSGTWLTY (Fig. 5a). Because patient samples were limited, we next studied the effect of mutations on epitope recognition by T cells from immunocompetent HLA-matched convalescent donors. We established peptide-stimulated T cell cultures and tested their reactivity against original and altered epitope peptides. In HLA-B*35:01-positive donor 1, stimulation with nucleocapsid TPSGTWLTY peptide, but no other HLA-B*35:01 candidates, expanded a viable T cell culture that strongly recognized TPSGTWLTY. The T332I exchange reduced T cell recognition, whereas subsequent T325K substitution fully abolished recognition (Fig. 5b). Thus, two sequential

mutations altering this epitope were required to eliminate T cell recognition.

T cells could also be expanded from two HLA-A*02:01-positive donors with peptide ALWEIQQVV from nsp8/ORF1ab, but none of the other six HLA-A*02:01 candidates (Supplementary Table 6). When these T cells were stimulated with 15-mer peptides encompassing ALWEIQQVV or its variant ALWEIQQFV (ORF1ab V4101F), only the original but not the variant was recognized (Fig. 5c). From two HLA-A*01:01-positive donors, T cell cultures were established by stimulation with mixed peptides CTDDNALAY and CTDDNALAYY from nsp9/ORF1ab. T cells from donor 2 recognized the nonameric peptide CTDDNALAY, and recognition of its variant was reduced (Fig. 5d). T cells from donor 3 responded to the decameric but not the nonameric peptide, and again mutation reduced recognition (Supplementary Fig. 2). Therefore, peptides CTDDNALAY(Y) may represent two T cell epitopes recognized by the same or different T cells. T cell responses to each of them are impaired by the ORF1ab T4164I mutation, which replaces an optimal anchor residue threonine in position 2 by suboptimal isoleucine.

To verify HLA restriction and test the impact of mutations on binding of HLA/peptide complexes to T cells, we stained T cell cultures with HLA/peptide tetramers. The results confirmed that TPSGTWLTY was HLA-B*35:01-restricted and CTDDNALAY(Y) were HLA-A*01:01-restricted (Fig. 5e). Cultures from donor 1 were dominated by CD8 T cells that bound to the HLA-B*35:01/TPSGTWLTY tetramer (77%), and consecutive mutation strongly reduced the proportion of tetramer-binding T cells down to 1.6% for the double mutant (Fig. 5e). Similarly, cultures from donor 2 contained substantial proportions of HLA-A*01:01/CTDDNALAY(Y) tetramer-binding CD8 T cells (18–23%), and mutant peptides produced a slight reduction in tetramer binding for the decameric epitope, and a strong reduction of tetramer-binding T cells for the nonameric epitope (Fig. 5f).

### High levels of CD8 T cells specific for the pre-mutant epitope

When we compared the timing of mutations in these CD8 T cell epitopes with the patient's absolute peripheral CD8 T cell numbers, we noted that there had been an increase in peripheral CD8 T cells immediately before mutation 1b (T325K) and mutation 3 (T4164I) arose (Supplementary Fig. 3). For the two early simultaneous mutations 1a and 2 (T332I, V4101F), the pattern was less clear due to a lack of

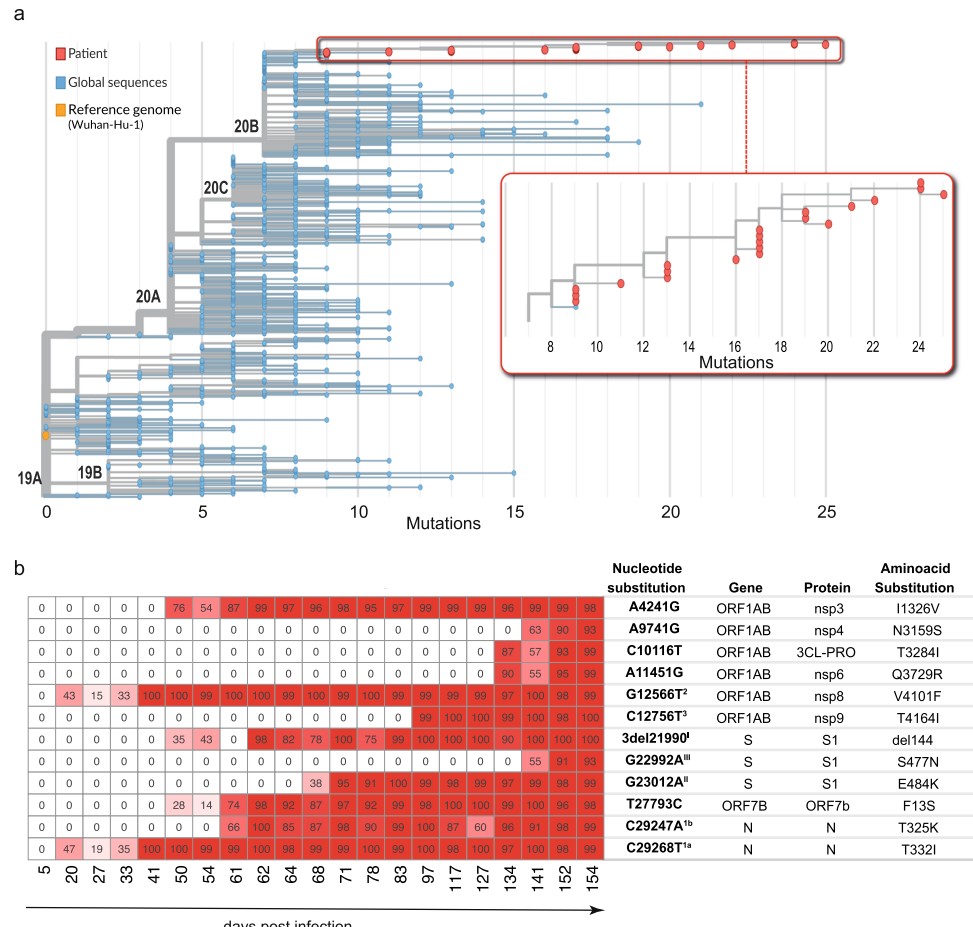

**Fig. 2 | Intrahost evolution of SARS-CoV-2. a** Phylogenetic tree of patient and global SARS-CoV-2 sequences. Maximum likelihood phylogenetic tree of the 21 whole genome SARS-CoV-2 sequences derived from the patient in this study (red) along with representative global sequences sampled between December 2019 and June 2020 (blue) in comparison to the reference genome Wuhan-Hu-1 (orange). A full list of global sequences with their GISAID accession numbers used in this analysis is given in the Supplementary Data File 1. The x-axis represents the genetic distance as nucleotide substitutions in relation to the reference genome. **b** Heatmap with frequencies of de novo SARS-CoV-2 mutations in this patient. Only non-synonymous mutations that became fixed in predominant variants are presented (see Supplementary Table 3 and Supplementary Fig. 2 for full list of mutations and sequence accession numbers). Nucleotide substitutions, gene and protein names, and amino acid substitutions are displayed. Superscript indicates potential immune escape mutations numbered as in Fig. 1.

measurements. Overall, fluctuations in CD8 T cell numbers in the patient were suggestive of CD8 T cell-driven immunoevasion.

A late-stage cryopreserved PBMC sample of the patient (day 145) was available for tetramer staining. While a large fraction (29.7%) of CD8 T cells bound to the wild type HLA-B*35:01 TPSGTWLTY multimer, binding was reduced (12.4%) with the single mutant peptide and nearly abolished (0.5%) with the double mutant (Fig. 6a–c). Thus, the patient had maintained high numbers of CD8 T cells specific for the original epitope 84 days after the second mutation, suggestive of strong cellular immune pressure. Two mutations were required to fully prevent these T cells from recognizing their target epitope.

## Discussion

Here we demonstrate the emergence of mutations that were associated with immune escape from antibodies and CD8 T cell responses in an immunocompromised individual with persistent, ultimately fatal COVID-19. Several previous case reports have described rapid viral diversification in the setting of insufficient immune control[34,35,48]. These studies were focused on the adaptation of the SARS-CoV-2 spike protein and identified several mutations located in neutralizing epitopes of the RBD and NTD. Strikingly, also in our case two mutations in the RBD and one in the NTD of spike emerged that have previously been shown in vitro to reduce neutralizing activity of antibodies targeting these regions[49–51]. Our data shows sensitivity of the patient's initial virus to neutralizing antibodies of transfused convalescent plasma that was abrogated by the subsequent mutant isolate. This further corroborates that these mutations resulted in escape against the transfused polyclonal antibodies in this immunocompromised host. The role of convalescent plasma in the management of immunocompromised COVID-19 patients with prolonged infection requires further investigation given its limited therapeutic efficacy and the risk of selection of immune escape variants[52].

Notably, the antibody escape mutations that evolved in this patient are also present in current or previous variants of concern/interest: E484K in Beta and Gamma, the deletion at position 144 in Alpha, and the mutation S477N in Omicron (BA.1, BA.2, BA.4, and BA.5)[53]. The independent emergence of such homoplastic mutations in separate individuals and their selection advantage to become predominant variants in different populations strengthens the hypothesis of convergent evolution in adaptation to the human immune system[30].

Our results suggest that selective pressure by CD8 T cells may contribute to shaping SARS-CoV-2 evolution in the setting of severe immunosuppression. In our case, two mutations associated with CD8 T cell escape occurred several weeks prior to spike mutations associated with antibody escape, and prior to administration of convalescent plasma. Subsequent antibody escape may have been facilitated by this

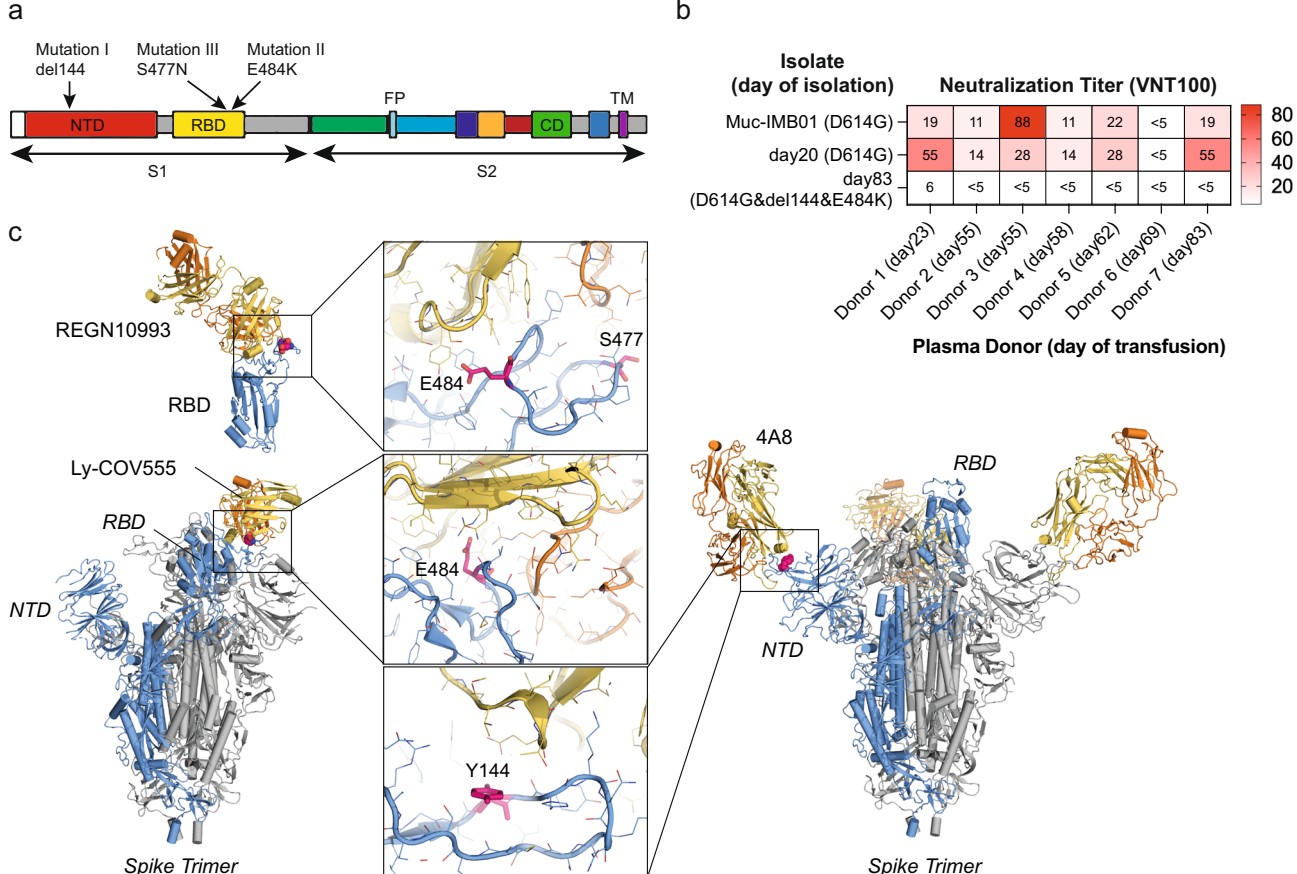

**Fig. 3 | Mutations in neutralising epitopes of the SARS-CoV-2 spike protein. a** Position of the three mutations in the spike gene coloured by domains. The deletion at position 144 (I) is located in the N-terminal domain (NTD) and S477N (III) and E484K (II) are located in the receptor binding domain (RBD). S1 Spike subunit 1, S2 Spike subunit 2, NTD N-terminal domain, RBD Receptor-binding domain, FP fusion peptide, CD connector domain; TM transmembrane domain. **b** Heatmap with virus neutralization titre (VNT100) corresponding to the reciprocal serum dilution that fully neutralized viral infection in vitro using Vero E6 cells. Neutralizing activity of serum samples taken from the donors of convalescent plasma on the day of donation was tested. The time point of each donor's plasma transfusion to the patient is indicated. Neutralizing responses were tested against an isolate from the

early pandemic phase only with the spike mutation D614G (Muc-IMB01) and autologous isolates of the patient. The time points of isolation of the autologous viruses and additional mutations in the spike protein to D614G are indicated. **c** Illustration of the three mutant amino acids using structures of SARS-CoV-2 spike domains in complex with antibodies REGN10993, Ly-COV555 and 4A8. Left and right subpanels: illustration of the positions of the mutation and binding of selected antibodies on the spike trimer. One spike protomer bound by an antibody is shown in blue, the other two in grey. Antigen-binding fragments (Fabs) or single-chain variable fragments (Fvs) are depicted in yellow (heavy chain) and orange (light chain). Middle subpanels: details of the antibody–spike protein interfaces, highlighting the roles of the amino acid changes in these interfaces in antibody escape.

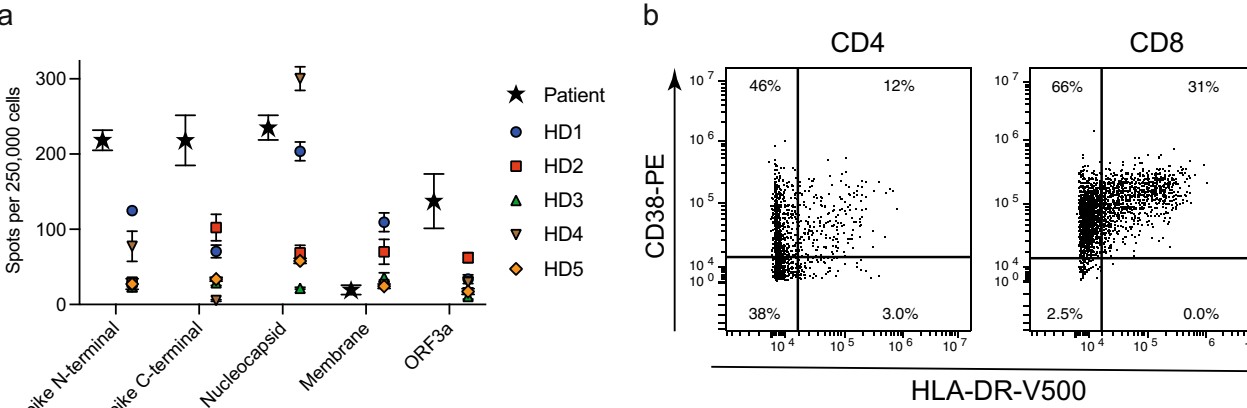

**Fig. 4 | T cell responses against SARS-CoV-2. a** T cell responses to selected SARS-CoV-2 antigens in the patient on day 127 after diagnosis and in five convalescent donors (HD1–5) after mild infection were tested in ex vivo IFN-γ ELISPOT using overlapping peptide pools spanning selected SARS-CoV-2 antigens as indicated

(lineage Wuhan-Hu-1). Tests were performed in three technical replicates. Mean values with standard deviation are shown. **b** Expression of activation markers CD38 and HLA-DR on the patient's peripheral CD8 or CD4 T cells were analyzed on day 145 by flow cytometry. Source data are provided in the Source Data file.

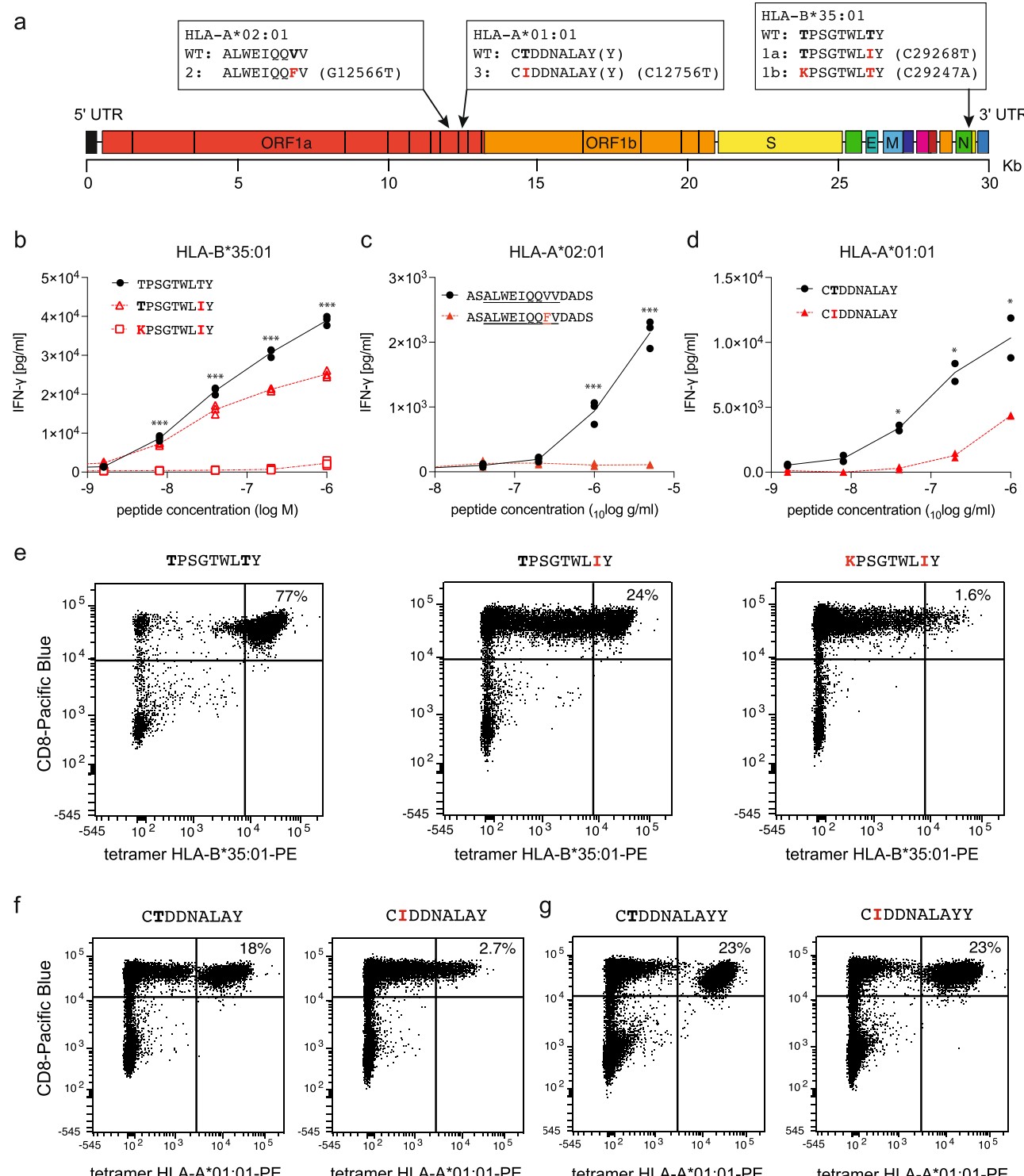

**Fig. 5 | Mutations in CD8 T cell epitopes result in immune escape. a** Locations of potential T cell epitopes in the SARS-CoV-2 genome. Sequences of original (WT) and mutant epitopes (1a, 1b, 2, 3) further studied below are shown in one-letter amino acid code with mutant residues in red. **b–d** Recognition of these three epitopes and their mutant versions by peptide-specific CD8 T cell cultures from convalescent donors was studied by overnight co-culture of T cells, HLA-matched activated B cells as antigen-presenting cells, and peptide at the indicated concentrations in triplicate. IFN-γ secreted into the supernatant was measured by ELISA. Individual data points are shown with the line representing the mean; *t*-test results are indicated as *$p < 0.05$, **$p < 0.01$ and ***$p < 0.001$ compared with the original peptide. T cell cultures were from donor 1 (**b**) and donor 2 (**c**, **d**). T cell cultures from donor 1 (**e**) or donor 2 (**f** + **g**) were stained with PE-labeled HLA-B*35:01 tetramers loaded with peptide TPSGTWLTY and its two mutant versions (**e**) or with HLA-A*01:01 tetramers loaded with CTDDNALAY(Y) and their mutant versions (**f** + **g**). Anti-CD8 antibody was labeled with Pacific Blue. Source data are provided in the Source Data file.

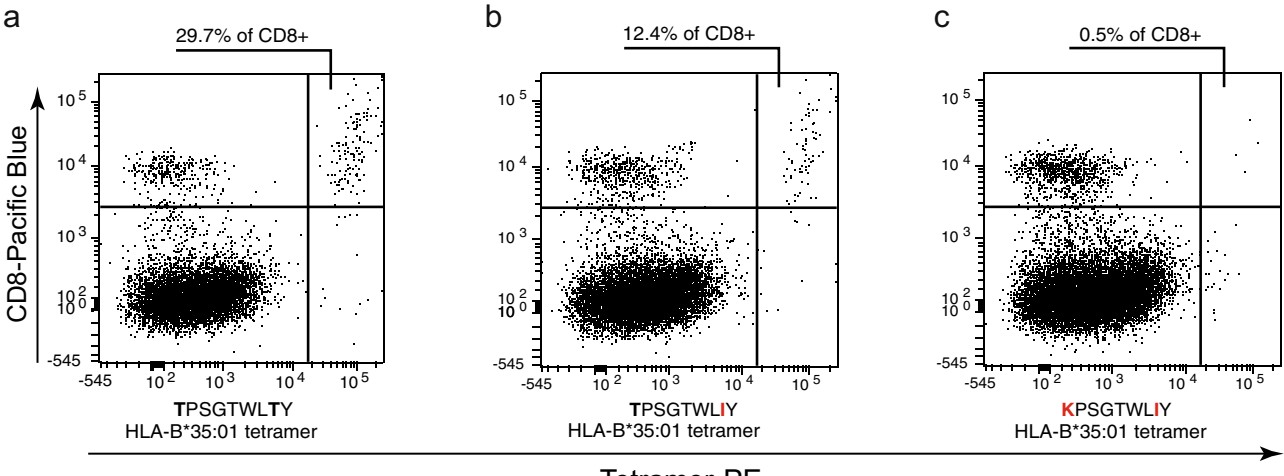

**Fig. 6 | Reduced recognition of a double-mutant CD8 T cell epitope by the patient's T cells. a–c** Peripheral blood cells from the patient at day 145 were stained with HLA-B*35:01/peptide tetramers loaded with the TPSGTWLTY peptide (**a**) or its two mutant versions (**b** + **c**).

jeopardised cellular immunity. It was proposed that mutations in SARS-CoV-2 variants have little impact on overall CD4 or CD8 T-cell reactivity in convalescents or vaccinees[5]. However, dozens of SARS-CoV-2 epitopes are recognized by average immunocompetent persons[8,54], whereas this number may be much smaller in immuno-compromised individuals due to a reduced T cell repertoire and functional T cell impairment[55,56]. Therefore, individual T cell epitopes and their variants may strongly affect control of infection in immunocompromised patients with reduced T-cell diversity, as observed for other complex viruses[57–59]. Evidence is now accumulating that CD8 T cell responses to ancestral SARS-CoV-2 are blunted by mutations occurring in certain viral variants[60–62]. It seems that in a subset of individuals (reported around 20% of subjects with a history of vaccination, infection or both) CD8 T cell recognition of omicron spike is reduced by more than 50%[63]. This may be highly relevant for future vaccination strategies.

Of note, most of the CD8 immune escape mutations observed in the present patient have independently emerged in multiple globally circulating lineages (Supplementary Table 7). However, CD8-evading mutations described here and elsewhere[62,64] have not become dominant in prevalent variants of concern so far, which may indicate that CD8 T cells exert stronger selective pressure in immunocompromised patients with defective B cell and antibody responses and chronic infection than in immunocompetent patients with a typical course of COVID-19. Furthermore, as opposed to antibody escape mutations in neutralizing epitopes of the SARS-CoV-2 spike protein that reduce antibody recognition in all hosts with antibodies targeting these regions, T cell escape mutations will only mediate immune evasion in hosts expressing the restricting MHC molecules and therefore do not confer a universal selection advantage for SARS-CoV-2.

Recent data has demonstrated the role of CD8 T cells in recovery from COVID-19 in patients with haematological malignancies[41]. However, the impact of CD8 T cells on virus evolution and immune evasion in immunocompromised patients has remained unclear. It is well established that strong CD8 T cell responses are directed to nucleo-protein epitopes including HLA-B*35:01/TPSGTWLTY[7,9,54,65]. CD8 T cell responses to nsp8/9/10 were less prominent in peptide-screening studies, in contrast to our results suggesting their substantial effect in this patient[7,54]. However, unbiased screening with intracellular expression fragments covering SARS-CoV-2 identified each of the epitopes HLA-A*02:01/ALWEIQQVV and HLA-A*01:01/CTDDNALAYY among the three most frequently detected CD8 T cell responses for their HLA restriction[65], consistent with a significant role of these

epitopes and the non-structural proteins nsp8 and nsp9 in T cell control of infection. CD8 T cells specific for epitopes from structural and non-structural proteins of a virus may functionally complement each other, since they recognize cells at different stages of infection[66]. In particular, CD8 T cells that recognize epitopes from regulatory proteins expressed early in virus-infected cells may be able to eliminate such cells before virions can be produced. In SARS-CoV-2, non-structural proteins that are expressed early, although not at high levels, strongly contribute to early peptide presentation on HLA class I[67].

There are some striking parallels between the present case and other recent findings. Stanevich et al have described the case of a patient with prolonged SARS-CoV-2 infection on the background of treatment with chemotherapy, rituximab and autologous stem cell transplantation for diffuse B cell lymphoma, who also received convalescent plasma therapy[60]. In that patient, like in our case, a number of virus mutations in sequences encoding potential CD8 T cell epitopes were detected that became dominant and fixed. T cell experiments showed that the corresponding amino acid substitution abolished recognition of the highest-scoring epitope by T cells from the patient. The affected epitope was the HLA-A*01:01-restricted peptide PTDNYITTY from nsp3. The same peptide was affected (in a different position) by mutation in our patient, who is HLA-A*01:01-positive, although we could not perform experiments on its T cell recognition due to a lack of patient material. A comprehensive screen across SARS-CoV-2 identified only eight HLA-A*01:01-restricted epitopes that were endogenously processed and presented to CD8 T cells and recognized by T cells from multiple donors[65]. In the aforementioned study two of these eight epitopes (PTDNYITTY and TTDPSFLGRY) were affected by one or more of a total of 34 fixed mutations of SARS-CoV-2[60]. In our case two epitopes were affected by two out of twelve mutations of SARS-CoV-2 (PTDNYITTY and CTDDNALAY(Y)). Considering that the nonredundant SARS-CoV-2 proteome has about 10,000 amino acids, it appears unlikely that such parallels are mere coincidence. Both cases support the hypothesis that CD8 T cells affect virus evolution in cancer patients with B-cell deficiency, but further investigations will be required to test this notion.

Our study has several limitations. It represents observations made in a single patient on limited numbers of samples from peripheral blood. Furthermore, additional unknown factors may have driven virus evolution. Therefore, detailed studies on a larger number of patients are required to prove a causal connection between immune pressure and virus evolution. Our analysis did not address potential CD4 T cell

evasion. We focused our analysis of T cell responses on CD8 T cells since activated CD8 T cells were predominant in the patient and cytotoxic CD8 T cell responses have previously been identified as major mediators of immune selection pressure in viral infections such as HIV, EBV and HCV[68–70]. In the meantime, multiple studies have also observed SARS-CoV-2 immunoevasion from CD8 T cell recognition[60–62,64].

In conclusion, we find that selection pressure shaping SARS-CoV-2 evolution may be mediated not only by antibody responses, but importantly also by CD8 T cell responses, in particular in patients with chronic infection and incomplete immunity.

## Methods

### Ethics declaration
This patient was part of the COVID-19 Registry of the LMU University Hospital Munich (CORKUM, WHO trial id DRKS00021225). Written informed consent was obtained before any study related procedure. All uses of human material have been approved by the LMU University Hospital Munich. Clinical and routine laboratory data was prospectively collected within the COVID-19 registry and verified and complemented by individual chart review. Patient data was pseudonymized for analysis and the study was approved by the local ethics committees (Ethikkommission der Medizinischen Fakultät der LMU München, Project No: 20-245). Healthy convalescent donors were recruited under ethical approval (Ethikkommission der Medizinischen Fakultät der LMU München, Project No: 17-455) of LMU University Hospital Munich and provided written consent.

### Sample collection
All five donors had acquired SARS-CoV-2 before November 2020 with a history of mild COVID-19 (WHO criteria). There were between the ages of 25 to 60 and two females and three males. Respiratory samples for viral cultures and SARS-CoV-2 sequencing were obtained from the patient by endotracheal aspiration and naso-pharyngeal swabs. Heparinized whole blood was collected for preparation of peripheral blood mononuclear cells (PBMCs) following standard procedures using Ficoll density centrifugation. PBMCs were stored over liquid nitrogen.

### Quantification of SARS-CoV-2 RNA from clinical specimens
The following PCR assays were used for quantification in the accredited routine diagnostics laboratory of the Max von Pettenkofer Institute as previously described[71]: The nucleocapsid (N1) reaction (Center for Disease Control (CDC) protocol (https://www.cdc.gov/coronavirus/2019-ncov/downloads/rt-pcr-panel-primer-probes.pdf accessed July-19-2022), see Supplementary Table 8), and the Roche Cobas SARS-CoV-2 nucleocapsid reaction (Roche, Mannheim, Germany). For nucleic acid extraction, the Maxwell RSC Viral Total Nucleic Acid Purification Kit was used with the Maxwell RSC-48 device (Promega GmbH, Fitchburg, USA). Quantification was based on standard curves using serial dilutions of two reference samples from INSTAND e.V. with $10^6$ and $10^7$ RNA copies per ml in duplicate.

### Viral culture and molecular detection
Vero E6 cells were inoculated at a dilution of 2:5 with an adsorption time of 1 h at 37 °C. The inoculum was removed, fresh cell culture medium was added [Modified Eagle medium (MEM) supplemented with 2% fetal calf serum, 1x MEM Non Essential Amino Acids solution and 5x penicillin/streptomycin and Fungizone™, all from Invitrogen, Thermo Fisher Scientific, Darmstadt, Germany] and cells were examined for cytopathic effect (CPE) every 24 h for 7 days. On day 0, when CPE was observed and, if no CPE was observed at the end of the cell culture incubation period, culture supernatants were collected and stored at −80 °C prior to nucleic acid extraction. Viral RNA was prepared from 140 μl of the cell culture supernatant stocks with the

QIAamp Viral RNA Mini Kit (Qiagen, Hilden, Germany) and eluted in 80 μl RNase free H2O. SARS-CoV-2 RNA from viral cultures was detected by RT-PCR using the primer/probe combinations of the WHO protocol[72]. Results of day 0 were compared to the results of the day when CPE was observed or of the end of the cell culture incubation period. The cell cultures were confirmed as positive if the CT-values of the RT-PCR was at least 3 CT-values lower compared to day 0. All cell cultures were performed within a class II cabinet in a biosafety laboratory level 3.

### SARS-CoV-2 serology
Patient sera samples were tested for IgG and IgA against the SARS-CoV-2 S1 subunit using a commercial ELISA-kit by Euroimmun (Lübeck, Germany) following the manufacturer's instructions. These assays were performed in the accredited routine diagnostics laboratories of the Max von Pettenkofer Institute and the Bundeswehr Institute of Microbiology, both Munich, Germany.

### Neutralisation test (NT)
Neutralising antibody titres were determined as previously described[73]. Neutralising activity of serum samples taken from the donors of convalescent plasma on the day of donation were tested against an early pandemic isolate (MUC IMB-1, GISAID accession number: EPI_ISL_406862) and expanded autologous patient isolates sampled on day 20 (DH206600, EPI_ISL_732538) and day 83 (DH209138, EPI_ISL_732531). Briefly, isolates were cultured in Vero E6 cells and NTs were performed in 96-well culture plates (Greiner bio-one, Frickenhausen, Germany). Virus stocks (50 TCID/50 μl) were prepared and stored at −80 °C until further use. Serum samples, including positive and negative control samples, were serially diluted in duplicates in Minimal Essential Medium (MEM, plus Non-Essential Amino Acids Solution and Antibiotic-Antimycotic Solution; all Invitrogen, Thermo Fisher Scientific, Darmstadt, Germany) starting at 1:5 to a maximum of 1:640. Diluted serum samples were pre-incubated with virus (1 h, 37 °C) before Vero E6 cells ($1 × 10^4$ cells/50 μl) were added to each well. After 72 h of incubation at 37 °C (5% $CO_2$) supernatants were discarded and the wells were fixed (13% formalin/PBS) and stained (0.1% crystal violet). The neutralising antibody titre corresponded to the reciprocal of the highest serum dilution showing complete inhibition of CPE. A virus retitration was performed in triplicate on each plate and exact titres were determined by retrograde calculation.

### NGS sequencing of SARS-CoV-2
The extracted RNA was translated into cDNA using the SuperScript IV First-Strand Synthesis System (Invitrogen, Thermo Fisher Scientific, Dreieich, Germany). After performing second strand synthesis (non-directional RNA second strand synthesis module NEBNext Ultra II, New England Biolabs, Frankfurt am Main, Germany), a library was generated using the Twist Library Preparation Kit (Twist Biosciences, San Francisco, CA, USA). Subsequently, a target enrichment step was added prior to sequencing on an Illumina MiSeq (Illumina Inc., Berlin, Germany). For this purpose, target enrichment baits that included SARS-CoV-2-specific sequences (Twist Respiratory Virus Research Panel, Twist BioSciences, San Francisco, CA, USA) were used according to the manufacturer's instructions. Captured libraries were sequenced using Sequencing V2 reagent chemistry with 2 × 150 cycles (Illumina Inc., Berlin, Germany). This approach was used for samples of days 20, 27, 33, 41, 50, 54, 62, 83, 97 and 154.

The samples of days 5, 61, 64, 68, 71, 78, 117, 127, 134, 141 and 152 were sequenced as amplicon pools generated from cDNA that was synthesised from isolated total RNA according to the ARTIC network nCoV-2019 sequencing protocol v2 https://www.protocols.io/view/ncov-2019-sequencing-protocol-v2-bp2l6n26rgqe/v2?version_warning=no. Amplicons were generated by two multiplex PCR

reactions with ARTIC v3 primer pools (IDT, USA, cat #10011442) pooled, diluted and quantified by Qubit DNA HS kit. Amplicon pools were diluted to 0.2 ng/µl and tagmented with Nextera XT Library Prep Kit (Illumina, USA). Nextera libraries were dual-barcoded and sequenced on an Illumina Hiseq 1500 instrument using V2 reagent chemistry and 2 × 75 cycles.

## Sequence analysis pipeline
Sequenced reads were demultiplexed and mapped against the SARS-CoV-2 reference genome (NC_045512.2) with BWA-MEM. Variants were called using Freebayes with a ploidy of 1[74]. In addition, the effects of genetic variants on amino acid sequences were analyzed with SnpEff[75]. Finally, the consensus sequences were created with the iVar package with default settings using the generated pileup files from samtools[76,77]. Generated consensus sequences were uploaded to the GISAID repository (Supplementary Table 3).

## Phylogenetic analysis and heatmap illustration
The phylogenetic tree was obtained from a local Nextstrain installation using the supplied Snakemake workflow[78]. Briefly, the workflow filters genomes based on pre-defined criteria, such as quality and lengths, aligns the genomes to the reference genome and constructs the phylogenetic tree, based on a maximum-likelihood approach. The sequences and metadata were downloaded from GISAID (accession date 09/06/2020) and subsampled to a maximum of 5 samples per country, year and month. The GISAID accession numbers of the used sequences are listed in the Supplementary Data file 1. The mutation frequency heatmaps were generated with the variant frequencies obtained from Freebayes using the R package pheatmap https://cran.r-project.org/package=pheatmap. Reported variants must show a minimal coverage of 10 reads and a variant frequency of at least 40% for coding variants in one of the samples.

## Structural analysis of mutant spike protein
Protein coordinates were obtained from the Protein Data Bank (www.rcsb.org), using the following accession codes: 6XDG, 7L3N, 7C2L. Structural analysis and figure preparation were done with PyMOL (Schrödinger, Munich, Germany). Previously characterised exemplary monoclonal antibodies were used to illustrate the interaction with neutralising epitopes in the RBD and NTD[79–81].

## HLA typing
HLA class I typing was performed to 4-digit resolution by accredited laboratories (Zentrum für Humangenetik und Laboratoriumsdiagnostik Dr. Klein, Dr. Rost und Kollegen, Martinsried, Germany, and Laboratory for Immunogenetics, LMU Klinikum, Munich, Germany).

## HLA class I epitope candidates
HLA class I epitope candidate peptides were identified by SAMBA, a simple anchor motif-based approach, considering all six of the patient's classical HLA class I allotypes (HLA-A*01:01, A*02:01, B*08:01, B*35:01, C*04:01, C*07:01)[82]. The anchor motifs used in this analysis are presented in Supplementary Table 5. Eight- to 10-mer peptides representing candidate epitopes and 15-mer peptides encompassing candidate epitopes were synthesised by JPT, Berlin, Germany (Research Track Plus peptide libraries).

## Cell culture
Standard medium for functional T cell analyses (ELISPOT, IFN-γ release) and T cell culture was RPMI 1640 with high glutamine (Gibco/Life Technologies, Paisley, United Kingdom), supplemented with 7.5% fetal calf serum (Anprotec, Bruckberg, Germany), penicillin (100 U, Gibco), streptomycin (100 µg/ml, Gibco), and sodium selenite (100 nM, ICN Biomedicals, Aurora, OH, USA).

## ELISPOT analyses
ELISPOT assays were performed with freshly purified PBMCs from patient and healthy convalescent donors. ELISPOT 96-well plates (PVDF membrane, Millipore MSIPN4510) were precoated overnight with an anti-IFN-γ antibody 1-D1K (15 µg/ml, Mabtech, Nacka, Sweden). Plates were washed, PBMCs were plated at 250,000 cells per well in standard medium, and peptide pools consisting of 15-amino acid peptides with 11 amino acids overlap fully covering SARS-CoV-2 antigens (Wuhan-Hu-1 strain; PepMix, JPT, Berlin, Germany) were added at 0.5 µg/ml per peptide to a total volume of 200 µl per well. Plates were incubated for 14–18 h at 37 °C under 5% carbon dioxide. Cells were removed, and plates were incubated with biotinylated anti-IFN-γ antibody 7-B6-1 (1 µg/ml; Mabtech) and streptavidin-alkaline phosphatase (1 µg/ml; Mabtech) according to the manufacturer's protocol. Spots were stained with AP conjugate kit (Bio-Rad, Puchheim, Germany) and counted in an automated ELISPOT counter (CTL, Bonn, Germany).

## T cell cultures and reactivity analyses
SARS-CoV-2 peptide-stimulated T cell cultures were set up in 24-well plates by plating cryopreserved, freshly thawed PBMCs from convalescent donors at 2.5 million PBMCs per ml of standard medium supplemented with interleukin-2 (10 U/ml, Proleukin S, Novartis, Nürnberg, Germany), with addition of individual peptides or mixes of up to five peptides at 0.5 µg/ml per peptide, in a total volume of 2 ml per well. After 5 ± 1, 10 ± 2 and 15 ± 2 days, cultures were maintained or expanded according to their visual appearance by either replacing half of the supernatant with fresh standard medium or by adding an equal volume of standard medium, in each case supplemented with interleukin-2 (100 U/ml). After 10 to 20 days, T cell cultures were used in tests of their reactivity to titrated peptides or in tetramer staining.

For peptide reactivity analysis of peptide-stimulated T cell cultures, T cells were thoroughly washed and incubated in 96-well culture plates in 3 or 4 replicates at 20,000 cells per well together with 10,000 CD40-stimulated B cells of appropriate HLA type and peptide in graded concentrations (from 0.0016 to 1 or 5 µg/ml in factor-5 steps) in standard medium (in a total of 200 µl/well)[83]. These co-cultures were incubated for 14 to 18 h, supernatants were harvested, and concentrations of IFN-γ were determined in a standard ELISA assay (Mabtech, Nacka, Sweden).

## HLA-peptide tetramer staining and flow cytometry
To confirm HLA-binding and T-cell recognition of the two predicted epitope candidates "TPSGTWLTY" and "CTDDNALAY(Y)", HLA-peptide tetramers were produced from HLA-A*01:01 or HLA-B*35:01 monomers (easYmer, ImmunAware, Copenhagen, Denmark) by incubation with peptide for two days at room temperature and subsequent tetramerization with streptavidin-PE (BD Biosciences, Heidelberg, Germany), according to the manufacturer's protocol. Cultured T cells or freshly thawn patient PBMCs were stained with tetramers at room temperature for 20 min, followed by staining with anti-CD8 Pacific Blue (HIT8a, cat. co. 300928, Biolegend, San Diego, CA, USA), anti-CD4 FITC (SK3, cat. no. 344604, Biolegend) and anti-CD3 A700 (SP34-2, cat. no. 557917, BD Biosciences) on ice for 20 min. Flow cytometry (FACS) was performed with a Becton-Dickinson Fortessa device, further data processing was done with FCSalyzer freeware (Sven Mostböck, Vienna, Austria). Lymphocytes were gated on FSC-A vs SSC-A (see Supplementary Fig. 4 for gating strategy).

Phenotypic FACS analysis was performed using fresh whole blood after lysis of erythrocytes. Activation markers on T cells were evaluated with a staining panel of anti-CD45 AF700 (clone HI30, Biolegend, catalogue number 304024) and anti-CD3 APC-H7 (clone SK7, BD Biosciences, cat. no. 347340), anti-CD4 PerCP (SK3, BD Biosciences, cat. no. 344624), anti-CD69 FITC (FN50, BD Biosciences, cat. no. 557049), anti-CD38 PE (HIT2, BD Biosciences, cat. no. 555460), and anti-HLA-DR V500 (G46-6, BD Biosciences, cat. no. 561224). See

Supplementary Table 9 for antibody details. The BD Multitest™ 6-color TBNK reagent with BD Trucount™ tubes (BD Biosciences) was used to determine the absolute counts of cells. Flow cytometry measurements were performed on a CytoFLEX LX flow cytometer (Beckman-Coulter). Data were analyzed with FCSalyzer (Sven Mostböck, Vienna). Lymphocytes were gated on FSC-A vs SSC-A and subsequently on CD4 vs. CD3 (see Supplementary Fig. 4 for gating strategy).

## Statistical analysis

IFN-γ secretion of T cell cultures was compared between stimulations with mutant and original peptides using $t$-tests. All calculations were performed using GraphPad Prism v.9.1.2 (GraphPad, San Diego, USA).

## Reporting summary

Further information on research design is available in the Nature Research Reporting Summary linked to this article.

## Data availability

All data are available within the Article and Supplementary files, or available from the corresponding authors on reasonable request. The SARS-CoV-2 sequence data used in this study is available at GISAID under the accession numbers listed in Supplementary Table 3 and Supplementary Data File 1. Raw reads have been deposited in NCBI GenBank under Bioproject ID PRJNA864063. Source data are provided with this paper.

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

## Acknowledgements

We thank the relatives of this patient for their support of this study. We greatly acknowledge all the colleagues working on the intensive care unit at the LMU Hospital in Großhadern Munich and the diagnostic teams at the Bundeswehr Institute of Microbiology (IMB) and the Pettenkofer-Institute, for their dedicated and excellent work in nursing and diagnostics of our patients. We acknowledge the iFlow Core Facility of the university hospital Munich (INST 409/225-1 FUGG) for assistance with the generation of flow cytometry data. This study was partially funded by the Medical Biological Defense Research Program of the Bundeswehr Medical Service (MHA, RW, AR, KM, PG, EG, SZ) and by LMUexcellent, funded by the Federal Ministry of Education and Research (BMBF) and the Free State of Bavaria under the Excellence Strategy of the Federal Government and the Länder, the FORCOVID (MM, OTK.) and BayVOC (MM, OTK) research initiatives. O.W. was supported by the Else-Kröner Excellence Fellowship from the Else-Kröner-Fresenius Stiftung (Project-ID 2021_EKES.13) and the Bavarian Center for Cancer Research (BZKF,

Lymphoma Study Group). O.W. was supported by the German Research Foundation (DFG)—Project no. 360372040-SFB 1335, P04) and BZKF (Bavarian Center for Cancer Research, Lymphoma Study Group). AM was supported by Wilhelm Sander-Stiftung (Project no. 2018.135.1).

## Author contributions

All authors contributed to the data collection, design of the study, writing of the manuscript and revision of the manuscript. E.K., M.M., A.M., M.A. conceived the study and wrote the manuscript. M.M., J.C.H, C.S. participated in clinical study design and data collection. A.G., H.B., S.K., M.H.A., Alexandra R. performed sequencing and bioinformatic analyses. K.-P.H. performed structural analysis. A.D. performed HLA analysis. S.G. contributed to convalescent plasma collection. P.G., K.M. performed neutralization assays. A.H., A.-W.M., A. L., Anna R., Enrico G., S.Z., A.M. performed laboratory analysis. E.K., J.C.H., Eric G., T.W., S.-S.S., O.W., H.-J.S. acquired patient data and samples and performed medical evaluation. M.S., O.T.K., M.v.B.-B., R.W. contributed resources. All authors read, edited and approved the manuscript.

## Funding

## Competing interests

The authors declare no competing interests.

## Additional information

Elham Khatamzas [1,2,3,14] ✉, Markus H. Antwerpen [4,5,14], Alexandra Rehn [4,5], Alexander Graf [6], Johannes Christian Hellmuth [1,3], Alexandra Hollaus [1,5], Anne-Wiebe Mohr [1,5], Erik Gaitzsch [1], Tobias Weiglein [1], Enrico Georgi [4,5], Clemens Scherer [3,7], Stephanie-Susanne Stecher [8], Stefanie Gruetzner [9], Helmut Blum [6], Stefan Krebs [6], Anna Reischer [1], Alexandra Leutbecher [1], Marion Subklewe [1], Andrea Dick [10], Sabine Zange [4,5], Philipp Girl [4,5], Katharina Müller [4,5], Oliver Weigert [1,11], Karl-Peter Hopfner [12], Hans-Joachim Stemmler [1], Michael von Bergwelt-Baildon [1,3,11], Oliver T. Keppler [3,5,13], Roman Wölfel [4,5], Maximilian Muenchhoff [3,5,13,14] & Andreas Moosmann [1,5,14]

[1]Department of Medicine III, University Hospital, Ludwig-Maximilians University Munich, Munich, Germany. [2]Division of Infectious Diseases and Tropical Medicine, Center for Infectious Diseases, Heidelberg Hospital, Heidelberg, Germany. [3]COVID-19 Registry of the LMU Munich (CORKUM), University Hospital, Ludwig-Maximilians University Munich, Munich, Germany. [4]Bundeswehr, Institute of Microbiology Munich, Munich, Germany. [5]German Center for Infection Research (DZIF), partner site Munich, Munich, Germany. [6]Laboratory for Functional Genome Analysis, Gene Center, Ludwig-Maximilians University Munich, Munich, Germany. [7]Department of Medicine I, University Hospital, Ludwig-Maximilians University Munich, Munich, Germany. [8]Department of Medicine II, University Hospital, Ludwig-Maximilians University Munich, Munich, Germany. [9]Institute for Transfusion Medicine and Haemostasis, Medical Faculty, University of Augsburg, Augsburg, Germany. [10]Laboratory for Immunogenetics, University of Munich, LMU, Munich, Germany. [11]German Cancer Consortium (DKTK), Munich, Germany. [12]Gene Center and Department of Biochemistry, Ludwig-Maximilians-Universität München, Munich, Germany. [13]Max von Pettenkofer Institute & Gene Center, Virology, Faculty of Medicine, Ludwig-Maximilians University, Munich, Germany. [14]These authors contributed equally: Elham Khatamzas, Markus H. Antwerpen, Maximilian Muenchhoff, Andreas Moosmann. ✉e-mail: elham.khatamzas@med.uni-heidelberg.de

