## [Peer Review File · Nature Communications]

Accumulation of mutations in antibody and CD8 T cell epitopes in a B-cell depleted lymphoma patient with chronic SARS-CoV-2 infectionREVIEWER COMMENTS

Reviewer #1 (COVID-19, convalescent therapy) (Remarks to the Author):

It is an interesting paper as it shows a longitudinal follow-up of a patients' COVID19 mutations. The story begins with the admission of a woman who is in her seventies. She experiences a severe chronic COVID19 infection. Her past medical history included follicular lymphoma and its therapy a month before her last admission.

1. Abstract: We ^{[[SEP]]}propose that human leukocyte antigen class I contributes to shaping the evolutionary landscape of SARS-CoV-2. ^{[[SEP]]}
The authors may better write this sentence after giving the data they have found about the HLA Class I alleles and CD8+ cells.

2. SARS-CoV-2 specific IgG and IgA were negative throughout the course of the disease. Is the antibody ELISA (Euroimmune) assay for the measurement of spike-specific antibodies? Did you have a chance to measure the serum immunoglobulin levels? They all may be at low levels. A regular protein antibody response is a T-cell-dependent response. The authors may add the lymphocyte numbers of the donor? Did they measure the lymphocyte subset levels of the patient? The patients' serum SARS-CoV-2 virus titers decreased (this might be contrary to the low neutralization titers), possibly by multiple doses of convalescent plasma. However, the tracheal titers stayed high (Suppl. Fig.1). This could be due to the lack of convalescent plasma-Ig transfer to the trachea. The patients' B cells are possibly low due to the anti-CD20 therapy and chemotherapy. So, this patient had a secondary immunodeficiency on admission affecting more severely the mucosa. Follicular lymphoma is frequently seen in COVID cases. As an immunologist, I could not suggest that she does not have a primary immunodeficiency.
Suppl Fig. 4 shows the reactivity to original and mutant epitope peptides, however, T cell culture from the patient is not compared with donor 3.

3. Suppl. Table 4. Could the authors explain the legend of this table, 'epitope candidates with sequence alterations by fixed SARS-CoV-2 mutations for all HLA class I alleles of the patient'?

^{[[SEP]]}

4. Suppl. Table 6 shows the mutations. Could the authors add a column in order to show the date of the mutation?

5. The authors suggested that the antibodies and CD8+ T cells drove SARS-CoV-2 evolution in chronic infection. The patient's CD8+ T cells have high reactivity to the SARS-CoV2. Do you think this might be related to the lymphopenia (?) instead of increased T cell response? Because the T cells are competent to eliminate the virus. However, the number might not be sufficient.

6. An IFN- γ ELISPOT assay that was performed with a fresh peripheral blood sample on day 127 showed that the patient had strong T-cell ^{[[SEP]]}responses to antigens spike (N-terminal and C-terminal half), nucleocapsid, and ORF3a. This increased T cell response maybe because of the insufficiency of B cells in the patient. The increase may be due to compensation. Could the authors comment on this in the discussion?

Reviewer #2 (SARS-CoV-2, viral immunity) (Remarks to the Author):

This manuscript provides a detailed virologic and immunologic characterization of a single immunocompromised B-cell depleted individual who experienced a prolonged SARS-CoV-2 infection. The authors demonstrate time-dependent intraindividual evolution of SARS-CoV-2 including the emergence of multiple antibody and CD8 T cell escape mutations that become fixed. They demonstrate that transfused convalescent plasma that neutralized the initial viral isolate was no longer able to neutralize antibody immune escape variants that developed during this prolonged infection. Additionally, they show that the virus developed three mutations in HLA-restricted CD8

epitopes that decreased CD8 T cell recognition by tetramer assays and reaction by IFN-gamma ELISA. Lastly, they similar to the antibody escape mutations, albeit to a lesser extent, the mutations in these CD8 epitopes are also found in naturally circulating SARS-CoV-2 viruses.

The mechanisms used by SARS-CoV-2 to escape from the adaptive immune system are not well understood. The majority of published reports have focused on evasion of neutralizing antibodies but multiple lines of evidence support an important role for T cells in SARS-CoV-2 protection. Thus understanding if, how and when SARS-CoV-2 evolves to avoid T cell recognition and how this impacts clinical disease is an important and broadly interesting area of research. This manuscript nicely documents that SARS-CoV-2 can evolve CD8 resistance during a prolonged infection in a single immunosuppressed individual. Their data, similar to other emerging data, suggests that CD8 escape may be occurring on a wider population level. The major limitation of this study is that it involves a single immunosuppressed individual which limits its generalizability and its ability to answer important questions like whether and how mutations in a limited number of CD8 epitopes affects viral control. Nevertheless, given the paucity of data in this area, this study is well done and will be of broad interest.

Minor recommendations:

General:

The authors should explicitly note in the discussion the limitations of this study

Figure 1A:

Suggest denoting in the figure each time the virus was sequenced. This would help the reader know in what interval the Ab- and CD8-escape mutations occurred.

Figure 2A:

The authors comment that individual T-cell epitopes and their variants may strongly affect control of infection in immunocompromised patients with reduced T-cell diversity. However, this patient seems to be able to broadly respond to multiple T-cell epitopes (responses tended to be stronger than those of five immunocompetent convalescent donors). If this is the case how important were mutations in a few CD8 (small percentage) T cell epitopes to this individual and are they likely to be important in a general non-immunosuppressed population?

Figure 2D and Supp Fig 4:

Recommend adding statistical comparisons.

Supp Figure 1:

Suggest adding the times at which each Ab- and CD8-escape mutation occurred to the figure to see if correlations exist with changes in viral load.

Supplementary Table 6:

Suggest noting if the Ab-escape and CD8-escape mutations are found in VOCs or VOIs.

Reviewer #3 (Viral immunity, T cell responses) (Remarks to the Author):

IN this study, the authors investigate SARS-CoV2 virus sequences and virus-specific T cells and antibodies within a single patient who had a malignancy and multiple immune dysfunctions and deficiencies. Further confounding the analysis is that the patient received multiple infusions of convalescent plasma from different donors, as well as monoclonal antibody therapy from Regeneron, and tested positive for infection for over 5 months preceding death.

The main conclusion which is the title of the study is that antibodies and CD8 T cells drive virus evolution. In this observational study of virus sequences over time, it is not possible to determine any causality or drivers of sequences-- only associations can be made. Moreover, the authors detect a robust T cell response throughout the infection, and yet, the virus continued to thrive, so it is not

possible to conclude that CD8 T cells had any real ability to clear or control the virus. Importantly, for the T cell response, no kinetic data are shown, so it is not possible to assess any evolution of the T cell response. While there were some dips in the magnitude of viral titers as shown in Supplementary Figure 1, the authors do not show any temporal relationship between a type of T cell response and subsequent appearance of a mutant virus that evades the response. There is no temporal association with the T cells response, which appears robust at day 145 as shown.

In conclusion, this study on a single individual has too many confounders and complexities that are not possible to control for. The large claims and conclusions made are not supported by the data.

“SARS-CoV-2 evolution and escape from CD8 T cells and antibodies during chronic infection” former title „CD8 T cells and antibodies drive SARS-CoV-2 evolution in chronic infection“

MS-No. NCOMMS-21-34564-T

Khatamzas et al

Point-to-point response to reviewer comments

Reviewer #1 (COVID-19, convalescent therapy) (Remarks to the Author):

It is an interesting paper as it shows a longitudinal follow-up of a patients' COVID19 mutations.

The story begins with the admission of a woman who is in her seventies. She experiences a severe chronic COVID19 infection. Her past medical history included follicular lymphoma and its therapy a month before her last admission.

Abstract: We propose that human leukocyte antigen class I contributes to shaping the evolutionary landscape of SARS-CoV-2.

The authors may better write this sentence after giving the data they have found about the HLA Class I alleles and CD8+ cells.

Response: Thank you for this suggestion to improve our wording. We have now rewritten the sentence as supported by our data: “CD8 T cell responses contribute to shaping the evolutionary landscape of SARS-CoV-2 and should be considered in future surveillance strategies.” (Lines 49-51)

SARS-CoV-2 specific IgG and IgA were negative throughout the course of the disease. Is the antibody ELISA (Euroimmune) assay for the measurement of spike-specific antibodies? Did you have a chance to measure the serum immunoglobulin levels? They all may be at low levels.

Response: Thank you for this great suggestion. We now specify more precisely which serological test was done: “...standard ELISAs (Euroimmune) for IgG and IgA against the S1 domain of the SARS-CoV-2 spike protein were negative throughout the course of disease.”

Total Immunoglobulin levels, namely IgA, IgG and IgM, were measured regularly during hospital admission and were below the normal range as already suggested in our immunosuppressed patient. We have added new Supplementary Tables 1 and 2 which show total antibody levels and lymphocyte subsets.

A regular protein antibody response is a T-cell-dependent response. The authors may add the lymphocyte numbers of the donor? Did they measure the lymphocyte subset levels of the patient?

Response: We have no information about the lymphocyte numbers of the respective healthy convalescent serum donors, but they had no underlying medical conditions, including no history of immunodeficiency. We now present the patient's lymphocyte subset levels in a new Supplementary Table 2 and selected T-cell subsets in Supplementary Fig. 6.

The patients' serum SARS-CoV-2 virus titers decreased (this might be contrary to the low neutralization titers), possibly by multiple doses of convalescent plasma. However, the tracheal titers stayed high (Suppl. Fig.1). This could be due to the lack of convalescent plasma-Ig transfer to the trachea. The patients' B cells are possibly low due to the anti-CD20 therapy and chemotherapy. So, this patient had a secondary immunodeficiency on admission affecting more severely the mucosa.

Follicular lymphoma is frequently seen in COVID cases. As an immunologist, I could not suggest that she does not have a primary immunodeficiency.

Response: On initial diagnosis of this patient with follicular lymphoma in our department neither investigations nor clinical history revealed any abnormalities suggestive of COVID. In retrospect it may be difficult to distinguish the contribution to the patient's immunodeficiency caused by the underlying lymphoma from the contribution of the three cycles of chemotherapy she received until shortly before SARS-CoV-2 infection.

With regards to the discrepancy of SARS-CoV-2 viremia and tracheal/bronchial viral load levels in this patient we suggest that this was multifactorial: SARS-CoV-2 RNAemia itself has been shown to be a marker for severe COVID-19 associated with higher mortality (Eberhardt et al PMID: 32962125, Fajnzylber et al PMID: 33127906, Rodriguez-Serrano et al PMID: 34162948), as seen in our patient. Data on the therapeutic efficacy of convalescent plasma are still controversial in particular in severe COVID-19 (Menicetti et al PMID: 34842924, Axford et al PMID: 34800996). Lack of antibody distribution to different compartments including the lung may be one of the factors accounting for this, animal data have so far failed to provide conclusive evidence. After convalescent plasma administration Deere et al (PMID: 34817208) showed no effect on SARS-CoV-2 viral titres in the non-human primate model whereas Takamatsu et al (PMID 34818068) describe in the syrian hamster model a significant reduction in viral titres in the lungs; this study was, however, limited to a observation period of 15 days under conditions of self-limiting infection.

We absolutely agree that this patient was severely immunocompromised both in peripheral blood and in mucosal sites. We believe this was a secondary immunodeficiency, since we have no evidence of an immunodeficiency prior to her lymphoma.

We made corresponding amendments to our text in lines 89-90, 166, 181-182.

Suppl Fig. 4 shows the reactivity to original and mutant epitope peptides, however, T cell culture from the patient is not compared with donor 3.

Response: Due to very limited availability of PBMC samples from the patient we are unable to provide T-cell culture data from the patient. We tried to set up T-cell cultures from a cryoconserved patient sample, but viability of the culture was not sufficient and T cells could not be tested. Therefore, we had to rely on T-cell cultures from HLA-matched convalescent donors as shown in Fig. 2d-f and Supplementary Figures.

3. Suppl. Table 4. Could the authors explain the legend of this table, 'epitope candidates with sequence alterations by fixed SARS-CoV-2 mutations for all HLA class I alleles of the patient'?

Response: Apologies for not being clear. We have now changed the title of this table to be more concise: “Putative CD8 T-cell epitopes affected by SARS-CoV-2 mutations in the patient”, and have added explanatory sentences to the notes below the Table.

4. Suppl. Table 6 shows the mutations. Could the authors add a column in order to show the date of the mutation?

Response: Thank you for this suggestion. We have now added a column to indicate the date of first detection of the mutation.

5. The authors suggested that the antibodies and CD8+ T cells drove SARS-CoV-2 evolution in chronic infection. The patient's CD8+ T cells have high reactivity to the SARS-CoV-2. Do you think this might be related to the lymphopenia (?) instead of increased T cell response? Because the T cells are competent to eliminate the virus. However, the number might not be sufficient.

Response: Thank you for raising these important points. The ELISPOT assay was performed with PBMC on day 127. In an analysis of absolute cell numbers at the closest available time point (day 145), the patient had normal numbers of monocytes, which are important as antigen-presenting cells in this assay, but there was lymphopenia with low numbers of lymphocytes and T cells and a lack of B cells. Therefore, the proportion in PBMCs of T cells specific for three SARS-CoV-2 antigens was higher in the patient than in the five convalescent donors, but this is unlikely to be true for the absolute number of SARS-CoV-2-specific T cells. We added information to the Results section (lines 178-185) to make this clearer. Unfortunately, we cannot make a direct quantitative comparison to specific T-cell numbers in the convalescent donors, since absolute T-cell numbers were not determined in these donors.

As to the question whether the lymphocyte numbers may have been sufficient to control infection, that is difficult to judge for us. Please be aware that the ELISPOT assay was performed with peptide pools representing non-mutated SARS-CoV-2 antigens (Wuhan-Hu-1 sequence), so it is possible that T-cell responses would have been reduced if we could have performed a similar assay with mutated peptide pools. We added this information to the text (line 178) and the legend of Fig. 2.

Whether or not CD8+ T cells will be competent enough to eliminate the virus (protective or non-protective) may depend on their antigen and even epitope specificity. Little is known on SARS-CoV-2 about this issue since it is difficult to study CD8+ T cells of different specificities separately in humans. This has been a long-standing, unresolved issue in other human viral infections where the antiviral CD8+ T-cell response has been studied for decades, e.g. CMV. For HIV, HLA associations and HLA footprints have allowed to identify CD8+ T cells, that are more likely to be protective. For SARS-CoV-2, we feel that the emergence of mutations in epitopes under selective pressure by T cells represents strong evidence for protectivity in CD8 T cells with these specificities – although we cannot rule out other unknown reasons that drove these mutations.

6. An IFN- γ ELISPOT assay that was performed with a fresh peripheral blood sample on day 127 showed that the patient had strong T-cell responses to antigens spike (N-terminal and C-terminal half), nucleocapsid, and ORF3a. This increased T cell response maybe because of the insufficiency of B cells in the patient. The increase may be due to compensation. Could the authors comment on this in the discussion?

Response: Thank you for these suggestions. As discussed above in response to your related point and clarified in our Results section, the increased SARS-CoV-2-specific T-cell response may be relative rather than absolute - we added explanatory text in lines 178-185. We agree that the importance of CD8 T cells compared to B cells and antibodies is probably elevated in this B-cell-deficient patient compared to typical SARS-CoV-2 infections in the larger population. If CD8 T cells would play a similarly dominant role in immune control of SARS-CoV-2 in the general population, we might expect to see more evidence of mutations in CD8+ T-cell epitopes in widespread viral variants than actually observed; however, a more diverse T-cell repertoire in immunocompetent persons that targets a larger number of epitopes, a diversified HLA and epitope repertoire in human populations, and a shorter course of infection that prevents emergence of such mutations may prevent emergence of dominant mutations clearly driven by CD8 T cells. We have added a remark to our discussion (lines 312-316) to address these points.

Reviewer #2 (SARS-CoV-2, viral immunity) (Remarks to the Author):

This manuscript provides a detailed virologic and immunologic characterization of a single immunocompromised B-cell depleted individual who experienced a prolonged SARS-CoV-2 infection. The authors demonstrate time-dependent intraindividual evolution of SARS-CoV-2 including the emergence of multiple antibody and CD8 T cell escape mutations that become fixed. They demonstrate that transfused convalescent plasma that neutralized the initial viral isolate was no longer able to neutralize antibody immune escape variants that developed during this prolonged infection. Additionally, they show that the virus developed three mutations in HLA-restricted CD8 epitopes that decreased CD8 T cell recognition by tetramer assays and reaction by IFN-gamma ELISA. Lastly, they similar to the antibody escape mutations, albeit to a lesser extent, the mutations in these CD8 epitopes are also found in naturally circulating SARS-CoV-2 viruses.

The mechanisms used by SARS-CoV-2 to escape from the adaptive immune system are not well understood. The majority of published reports have focused on evasion of neutralizing antibodies but multiple lines of evidence support an important role for T cells in SARS-CoV-2 protection. Thus understanding if, how and when SARS-CoV-2 evolves to avoid T cell recognition and how this impacts clinical disease is an important and broadly interesting area of research. This manuscript nicely documents that SARS-CoV-2 can evolve CD8 resistance during a prolonged infection in a single immunosuppressed individual. Their data, similar to other emerging data, suggests that CD8 escape may be occurring on a wider population level. The major limitation of this study is that it involves a single immunosuppressed individual which limits its generalizability and its ability to answer important questions like whether and how mutations in a limited number of CD8 epitopes affects viral control. Nevertheless, given the paucity of data in this area, this study is well done and will be of broad interest.

Minor recommendations:

General:

The authors should explicitly note in the discussion the limitations of this study

Response: We have added a paragraph with an explicit discussion of limitations of this study, including the Reviewer's important point that we should be careful to generalize our findings from a single patient (lines 338-341).

Figure 1A:

Suggest denoting in the figure each time the virus was sequenced. This would help the reader know in what interval the Ab- and CD8-escape mutations occurred.

Response: We have added a "DNA" symbol in Fig. 1A for each time point when virus was sequenced.

Figure 2A:

The authors comment that individual T-cell epitopes and their variants may strongly affect control of infection in immunocompromised patients with reduced T-cell diversity. However, this patient seems to be able to broadly respond to multiple T-cell epitopes (responses tended to be stronger than those of five immunocompetent convalescent donors). If this is the case how important were mutations in a few CD8 (small percentage) T cell epitopes to this individual and are they likely to be important in a general non-immunosuppressed population?

Response: We agree that it seems contradictory that the patient reacted quite strongly to antigens spike, ORF3a, and nucleocapsid in the ELISPOT assay on day 127. Reviewer 1 raised our attention to the fact that we have not described this result with sufficient precision. While T-cell responses to these antigens were, on average, stronger than responses of convalescent donors on a per-cell basis, it is less likely that these responses were stronger than those of convalescent donors on an absolute basis. The reason for this is that PBMC and T-cell numbers were much lower in the patient than in normal donors, and therefore the 250,000 PBMCs used in each well of the assay represent the cells from a larger volume of blood of the patient than of the convalescent donors. Unfortunately, we cannot present precise calculations, since absolute PBMC numbers of the healthy donors were not determined. We have amended our results section (lines 178-185) to make this situation clearer.

The ELISPOT assay in Fig. 2A was performed with antigen-covering peptide pools, and therefore it is difficult to draw conclusions on the diversity of epitope specificities for each antigen. These commercial peptide pools corresponded to original SARS-CoV-2 antigens from the Wuhan-Hu-1 variant. We have clarified this in line 186 and the legend of Fig. 2. Mutant peptides might have strongly reduced the signals. A minimal interpretation of this experiment would be that at least one T-cell epitope recognized by the patient's T cells was present in each of S1, S2, ORF3a, and N, whereas non-synonymous, fixed SARS-CoV-2 mutations were identified in the patient in S1 and N, but not S2 and ORF3a.

Why were there no potentially immune-evading mutations in S2 and ORF3a? A hypothetical answer may be that S2 and ORF3a tend to elicit CD4 rather than CD8 T-cell responses (Cohen et al., 2021). In contrast, N tends to induce CD8 T-cell responses, and cytotoxic CD8

T cells may have a more direct role in control of SARS-CoV-2 than CD4 T cells, especially in the absence of B-cell responses. It seems likely to us that the T-cell response to N was strongly dominated by CD8 T cells specific for a single epitope, TPSGTWLT_Y in its non-mutant form, since (1) patient multimer staining on day 145 identified close to 30% of all peripheral CD8 T cells as specific for this epitope, a very high number that might well explain the majority of the N-specific ELISPOT response, and (2) since two out of two non-synonymous mutations in N were precisely located within this epitope, a phenomenon unlikely to have emerged by chance.

**Figure 2D and Supp Fig 4:
Recommend adding statistical comparisons.**

Response: We assume that the reviewer refers to an apparent lack of error bars in Figure 2D and Sup. Fig. 4. However, these diagrams did include error bars, but for most data points the error margins were too small for the error bars to become visible. We have changed the diagrams to make the error bars more clearly visible. Moreover, we now show the results from t tests in the form of asterisks for these T-cell experiments.

**Supp Figure 1:
Suggest adding the times at which each Ab- and CD8-escape mutation occurred to the figure to see if correlations exist with changes in viral load.**

Response: We have revised Sup. Fig. 1, adding symbols to indicate the time points of mutations.

**Supplementary Table 6:
Suggest noting if the Ab-escape and CD8-escape mutations are found in VOCs or VOIs.**

Response: We have revised Supplementary Table 6 (now Supplementary Table 8) to indicate whether mutations are present in VOCs or VOIs.

Reviewer #3 (Viral immunity, T cell responses) (Remarks to the Author):

IN this study, the authors investigate SARS-CoV2 virus sequences and virus-specific T cells and antibodies within a single patient who had a malignancy and multiple immune dysfunctions and deficiencies. Further confounding the analysis is that the patient received multiple infusions of convalescent plasma from different donors, as well as monoclonal antibody therapy from Regeneron, and tested positive for infection for over 5 months preceding death.

Response: We fully concur with the reviewer that conclusions from studies on single patients should be drawn with sufficient care.

We have added a section that presents this and other limitations of our study to the discussion, lines 338-341.

However, we would also like to point out that important hypothesis-generating observations and findings have and can be made with single patients. On the topic of SARS-CoV-2, single case studies on antibody immune evasion in spike have been published very prominently, although immune evasion in spike was much more expected than CD8 T-cell evasion, which has been studied comparatively little so far.

Our study provides evidence for combined evasion of SARS-CoV-2 from antibody and CD8 T-cell responses in an immunosuppressed patient who had SARS-CoV-2-specific CD8 T-cells, but no B cells, and received convalescent plasma specific for parental SARS-CoV-2. The patient did not receive antibodies from Regeneron. We feel that evidence for joint action of antibodies and CD8 T-cells in driving its immune evasion by mutation is not suggestive of confounders but is in accordance with accepted principles of antiviral immunity. Immune responses to viruses typically consist of antibody and T-cell responses that are supposed to jointly contribute to control of infection.

The main conclusion which is the title of the study is that antibodies and CD8 T cells drive virus evolution. In this observational study of virus sequences over time, it is not possible to determine any causality or drivers of sequences-- only associations can be made.

We agree that demonstration of causality is difficult. We have changed our title, eliminating any direct reference to causality between immune responses and viral evolution.

Moreover, we have eliminated claims of such causality throughout the text (lines 43, 298-299, 300, 311, 325, 346), and have added a paragraph on limitations of our study to the Discussion.

Moreover, the authors detect a robust T cell response throughout the infection, and yet, the virus continued to thrive, so it is not possible to conclude that CD8 T cells had any real ability to clear or control the virus.

Response: While we concede not having demonstrated causality between immune responses and virus evolution, we have a different view regarding the strength of our evidence. In particular, we feel that the continuous maintenance of strong CD8 T-cell responses capable of recognizing non-mutant but not mutant epitopes, together with full suppression of virus carrying non-mutant epitopes, constitutes (circumstantial) evidence in favour of, not against, the ability of CD8 T cells to control non-mutant virus. As shown in Fig. 1b in connection with the data in Fig. 2, the virus that continued to thrive was fully mutated in several CD8 T-cell epitopes (mutations T332I, V4101F, T325K, T4164 reached >98% prevalence). Virus without these mutations was completely eliminated during the patient's disease.

Further evidence for an active role of CD8 T cells was also suggested by an early bronchial lavage sample (day 14) that showed the presence of T cells, with 75% of them CD8-positive. As no further time points of immunophenotyping of bronchial specimens were available, we decided not to include this information in the manuscript.

Importantly, for the T cell response, no kinetic data are shown, so it is not possible to assess any evolution of the T cell response. While there were some dips in the magnitude of viral titers as shown in Supplementary Figure 1, the authors do not show any temporal

relationship between a type of T cell response and subsequent appearance of a mutant virus that evades the response. There is no temporal association with the T cells response, which appears robust at day 145 as shown.

Response: We agree that kinetic data on SARS-CoV-2-specific T-cell responses would have been desirable, but since patient T-cell samples from earlier time points are not available, we regret that this avenue is closed to us.

We hope we can partially address the reviewer's point by presenting all available data on T-cell subsets in peripheral blood (new Suppl. Table 2, new Suppl. Fig. 6). Suppl. Fig. 6 suggests that CD8 T cells may have increased at time points preceding key mutations in CD8 epitopes. For mutation T325K (mutation 2b), available data allow to discern that total and activated CD8 T-cells increased during the days leading up to the mutation and decreased thereafter, which is what would be expected if T cells specific for the non-mutant (T325) epitope exerted selective pressure on the virus.

That said, replacement of a non-mutant by a mutant virus through T-cell action might in some cases be a gradual process that may require several weeks to complete. See, for example, in Fig. 1b the N T332I (mutation 1a) and nsp8 V4101F (mutation 2) mutations that took at least 21 days from first detection to complete dominance. The gradual nature of these replacements, and the limited number of time points available in Suppl. Fig. 6, may explain why a T-cell peak was not detected simultaneous to these mutations.

In conclusion, this study on a single individual has too many confounders and complexities that are not possible to control for. The large claims and conclusions made are not supported by the data.

Response: We have moderated our claims and conclusions, in particular with reference to causality, in our title and text, and have added a paragraph on limitations of our study.

REVIEWER COMMENTS

Reviewer #1 (Remarks to the Author):

Attached as a file.

Reviewer #2 (Remarks to the Author):

The authors have adequately addressed all of my concerns and comments.

Reviewer #3 (Remarks to the Author):

The authors provided some additional analyses to address the comments. The major conclusions, that the virus mutates for antibody and CD8 T cell escape, in my opinion, is still not supported by the data presented. In particular, the conclusions that the virus evolves to evade CD8 t cell responses in the context of immunosuppression are not supported by the data. Notably, the patient has a much higher CD8 T cell response against peptide pools of viral epitopes compared to non-immunosuppressed convalescent donors, and the patients CD8 T cells also respond to several peptide epitopes in the mutant virus (Fig 2g) and there is a reduced response to one of the mutant epitopes. However, the CD8 t cell response overall is highly activated (Fig. 2b), and in vivo, the response is to multiple epitopes, so the conclusion that the virus is evading responses by CD8 T cells is incorrect. Moreover, there is scant analysis of the CD8 T cell response at other timepoints-- the data presented show only results from a late (day 145) timepoint which are quite robust. The authors would need to show that there is a net reduction of the CD8 T cell response over time to support their conclusion that the virus is evading T cell recognition. But they don't have these data, and there is no evidence that this phenomenon is occurring, so the bottom line of the study, including the title is not supported by the data. There may be multipel factors contributing to the persistence of virus in this complex patient, including that the CD8 t cell are functionally impaired, that the virus had infiltrated other tissues, and other complicating features of the underlying lymphoma and the fact that she had very few B cells from previous treatments.

Thanks to the authors for the efforts to improve the manuscript.

The authors gave the HLA Class I alleles they found in the patient as 'HLA- A*01:01, A*02:01, B*08:01, B*35:01, C*04:01, C*07:01.

And they wrote on 'HLA-peptide tetramer staining and flow cytometry' as given below:

HLA-peptide tetramers were produced from HLA-A*01:01 or HLA-B*35:01 monomers (easYmer, ImmunAware) by incubation with peptide for two days at room temperature and subsequent tetramerization with streptavidin-PE (Becton-Dickinson), according to the manufacturer's protocol. Cultured T cells or freshly thawed patient PBMCs were stained with tetramers at room temperature for 20 min, followed by staining with anti-CD8 Pacific Blue, anti-CD4 FITC (both Biolegend) and anti-CD3 A700 (BD Pharmingen) on ice for 20 min. Flow cytometry (FACS) was performed with a Becton-Dickinson Fortessa device, further data processing was done with FCSalyzer freeware (Sven Mostböck, Vienna).

Phenotypic FACS analysis was performed using fresh whole blood after lysis of erythrocytes. Activation markers on T cells were evaluated with a staining panel of anti-CD45 AF700 (Biolegend) and anti-CD3 APC-H7, anti-CD4 PerCP, anti-CD69 FITC, anti-CD38 PE, and anti- HLA-DR V500 (BD Biosciences). The BD Multitest™ 6-color TBNK reagent with BD Trucount™ tubes (BD Biosciences) was used to determine the absolute counts of cells. Flow cytometry measurements were performed on a CytoFLEX LX flow cytometer (Beckman- Coulter). Data were analyzed with FCSalyzer (Sven Mostböck, Vienna).

Questions regarding HLA dependent analyses:

- Authors suggests the HLA Class I alleles contribution on shaping the evolutionary landscape of SARS-CoV-2. Why did they use only HLA-A*01:01 and HLA-B*35:01 monomers? Didn't the authors use HLA-A*02:01, HLA-B*08:01, and C alleles during the staining and flow cytometry tests?

SARS-CoV-2 specific IgG and IgA were negative throughout the course of the disease. They suggested that they tested for IgG and IgA against the S1 domain of the SARS-CoV-2 spike protein. And they added that they were negative throughout the course of disease. They are given as below in suppl. table. They were very low (lowest level: 2.11 g/l (reference: 7-16 g/l)), not only below normal range. They possibly varied due to the convalescent plasma recurrently given to the patient.

They added in the manuscript that IgG and IgA against the S1 domain of the SARS-CoV-2 spike protein were negative throughout the course of disease. How can the authors wait it to be positive when their patients' antibody levels are as low as an agammaglobulinemic (generally, 1.0-2.0 g/l) patient. In such a patient with low IgG, the evaluation of the specific antibodies does not make sense.

Supplementary Table 1: Immunoglobulin levels measured over time in the patient as part of clinical routine care.

	Reference range	day 2	day 10	day 17	day 26	day 39	day 52	day 59	day 68	day 81	day 88	day 95
IgG	7.00 - 16.00 (g/l)	4.11	3.02	3.13	2.11	2.46	2.13	3.1	4.1	3.71	4.33	4.2
IgA	0.70 - 4.00 (g/l)	1.08	0.86	0.61	0.52	0.58	0.53	0.79	0.97	0.78	0.95	0.88
IgM	0.40 - 2.30 (g/l)	0.13	0.11	0.1	0.12	0.1	0.14	0.15	0.18	0.19	0.3	0.25

Although the authors, in the new title, 'SARS-CoV-2 evolution and escape from CD8 T cells and antibodies during chronic infection', better defined the mechanism of evolution and escape during the chronic infection in this immunodeficient patient, the emphasis on only CD8+ T cells could not be understood as we saw, in this patient, the results of the defect in protein antibody response, which is very much related with CD4+ T cell levels. So, the emphasis on HLA Class I alleles and CD8+ T cells in this manuscript did not make sense. The authors may better think on covering this important issue while they are revising their manuscript.

Note: I could not see in the methods the details of the measurement of antibodies against the S1 domain of the SARS-CoV-2 spike protein. They may better add it to the methods to make things clearer.

I thank the authors for adding the lymphocyte subset levels. It is clearly seen in the Table that there is severe lymphopenia, ranging from 74 and 452 per \$\mu\$ l (1220-3560). In this situation, not only number of CD8+ T cells nor CD4+ T cells is sufficient to show an effect on SARS-CoV-2.

In previous evaluation, the authors possibly misunderstood my comment on lymphocyte subsets. The lymphocyte subset asked below was not the lymphocyte subset of the healthy convalescent plasma donor.

- Question: A regular protein antibody response is a T-cell-dependent response. The authors may add the lymphocyte numbers of the donor? Did they measure the lymphocyte subset levels of the patient?
- Response: We have no information about the lymphocyte numbers of the respective healthy convalescent serum donors, but they had no underlying medical conditions, including no history of immunodeficiency. We now present the patient's lymphocyte subset levels in a new Supplementary Table 2 and selected T-cell subsets in Supplementary Fig. 6.

Note: They wrote convalescent serum in the manuscript. Did they use serum or plasma for the patient?

Supplementary Table 2. Summary of lymphocyte subsets measured over time in the patient as part of routine clinical care.

	Reference Range	day 5	day 28	day 47	day 54	day 61	day 68	day 75	day 82	day 89	day 93	day 96
Lymphocytes absolute	1220-3560	74	121	266	170	266	264	326	452	315	252	452
% Leucocytes	18-46%	<1	2	1	1	2	2	2	4	2	4	7
T cells absolute (CD45+ CD3+)	620-2020	32	71	76	58	89	58	69	180	231	89	233
% Lymphocytes	60-85 %	43	59	29	34	31	22	21	4	41	35	52
CD4+ T cells absolute (CD45+ CD3+ CD4+ CD8-)	380-1300	9	31	37	18	28	25	34	131	97	56	168
% Lymphocytes	31-62 %	12	26	14	10	10	10	11	29	31	22	37
CD8+ T cells absolute (CD45+ CD3+ CD4- CD8+)	160-810	22	37	37	38	57	33	31	44	30	30	59
% Lymphocytes	14-43%	30	31	14	22	20	12	9	10	9	12	139
CD38+ CD8 T cells absolute (CD45+ CD3+ CD4- CD8+ CD38+)	< 210	22	36	36	36	56	30	30	37	25	29	56
% Lymphocytes	< 7%	30	30	14	21	20	11	9	8	8	12	12
HLA DR CD8 T cells absolute (CD45+ CD3+ CD4- CD8+HLADR+)	< 124	19	31	25	30	40	18	16	19	16	17	39
% Lymphocytes	< 62%	84	83	68	79	71	54	53	43	65	59	66
CD4/CD8 Ratio	0.9-3.9	0.4	0.83	1	0.46	0.53	0.79	1.21	3.19	3.25	1.93	2.82
CD16+CD56+T cells absolute (CD45+ CD3+ CD16&56+)	< 210	2	4	5	5	11	3	3	14	9	2	9
% Lymphocytes	< 13%	3	3	2	2	3	1	1	3	2	8	2
B cells absolute (CD45+ CD19+)	70-420	0	0	0	0	0	0	0	0	0	0	0
% Lymphocytes	6-20 %	<1	0	0	0	0	0	0	0	0	0	0
NK cells absolute (CD45+ CD16&56+ CD3-)	50-510	42	50	190	113	195	208	256	272	186	161	222
% Lymphocytes	4-30%	56	42	71	67	68	79	79	60	59	64	49

The authors suggested that SARS-CoV-2 RNAemia itself has been shown to be a marker for severe COVID-19 associated with higher mortality. However the reason of SARS-CoV-2 RNAemia was possibly derived from the insufficient response of the immune system.

They suggest that the data on the therapeutic efficacy of convalescent plasma are still controversial in particular in severe COVID-19. However, in patients with antibody deficiency its effect has been clearly demonstrated and the guidelines regarding SARS-CoV-2 management for patients with inborn errors of immunity (IEI) have already included this therapy. Given the presented patient has a severe hypogammaglobulinemia, she might have benefited if the mutant virus experienced by the donors was the same as the patient.

There are other questions to be answered. The study evaluated a single immunosuppressed individual. Evolution of the SARS-CoV-2 in this study seems to be related to the incomplete cellular and humoral response of this specific patient. It is difficult to suggest that the virus may evolve in a such a fast way in an individual with a normal immune response involving specific antibody and specific T cells. This issue may be discussed in the discussion.

Minor comments:

Suppl. Figure 1A: The 1a, 1b, 2 and 3 should be defined in the figure legend. Moreover, the day that the antiviral therapy, IVIG, convalescent plasma was received by the patient should be indicated.

Supplementary Figure 2 is very demonstrative to show the date of the mutations. Did the authors see any correlation with the therapies the patient received?

Supplementary Fig 4: The original and mutant epitopes are given from donor 3. However, I could not understand which one is original and which one is mutant from the table.

Supplementary Figure 5: What did the authors expect when they use HLA-B*35:01 tetramer for the donor 1 and HLA-A*01:01 for donor 2? What was their conclusion?

Page 3, Line 79: Chronic SARS-CoV-2 infection in a patient after ?immunosuppressive? cancer therapy.

The authors should not specify the cancer therapy as immunosuppressive. All chemotherapies leads to immunosuppression. Instead they could indicate the time passed after therapy.

“SARS-CoV-2 evolution and escape from CD8 T cells and antibodies during chronic infection”

MS-No. NCOMMS-21-34564-A

Khatamzas et al

Point-to-point response to reviewer comments

REVIEWER COMMENTS

Reviewer #1 (Remarks to the Author) - copied from pdf file:

Thanks to the authors for the efforts to improve the manuscript.

The authors gave the HLA Class I alleles they found in the patient as ‘HLA- A*01:01, A*02:01, B*08:01, B*35:01, C*04:01, C*07:01.

And they wrote on ‘HLA-peptide tetramer staining and flow cytometry’ as given below:

HLA-peptide tetramers were produced from HLA-A*01:01 or HLA-B*35:01 monomers (easYmer, ImmunAware) by incubation with peptide for two days at room temperature and subsequent tetramerization with streptavidin-PE (Becton-Dickinson), according to the manufacturer’s protocol. Cultured T cells or freshly thawed patient PBMCs were stained with tetramers at room temperature for 20 min, followed by staining with anti-CD8 Pacific Blue, anti-CD4 FITC (both Biolegend) and anti-CD3 A700 (BD Pharmingen) on ice for 20min. Flow cytometry (FACS) was performed with a Becton-Dickinson Fortessa device, further data processing was done with FCS analyser freeware (Sven Mostböck , Vienna).

Phenotypic FACS analysis was performed using fresh whole blood after lysis of erythrocytes. Activation markers on T cells were evaluated with a staining panel of anti-CD45 AF700 (Biolegend) and anti-CD3 APC-H7, anti-CD4 PerCP, anti-CD69 FITC, anti-CD38 PE, and anti-HLA-DR V500 (BD Biosciences). The BD Multitest™ 6-color TBNK reagent with Trucount™ tubes (BD Biosciences) was used to determine the absolute counts of cells. Flow cytometry measurements were performed on a Cyto FLEX LX flow cytometer (Beckman-Coulter). Data were analysed with FCS analyzer (Sven Mostböck , Vienna).

Questions regarding HLA dependent analyses:

- Authors suggests the HLA Class I alleles contribution on shaping the evolutionary landscape of SARS-CoV-2. Why did they use only HLA-A*01:01 and HLA- *35: 1 monomers? Didn’t the authors use HLA-A*02:01, HLA-B*08:01, and C alleles during the staining and flow cytometry tests?

Response: We apologise for the confusion created here. We did indeed consider the full HLA-class I haplotype of this patient (HLA- A*01:01, A*02:01, B*08:01, B*35:01, C*04:01, C*07:01) when we examined potential immune selection pressure mediated by CD8 T-cells. However, our analysis was not done primarily by tetramer staining, but by studying the genomic regions where fixed mutations arose to check for CD8 T-cell epitope candidates using an anchor-motif based approach as outlined in the Supplement on page

4 (paragraph: “HLA class I epitope candidates”) and Supplementary Table S3. The full set of peptides was synthesized, and in the next step peptide-stimulated T cell cultures were set up from HLA-matched SARS-CoV-2-reconvalescent donors. It was then tested in functional assays such as those shown in Fig. 2 d-f whether peptide-specific T cells had been obtained.

Tetramer staining was used in a last step in order to confirm the predicted HLA class I restriction, as shown in Supplementary Fig. 5 and in the main Fig. 2g. These were the epitope candidates “TPSGTWLTY” and “CTDDNALAY(Y)”, which we confirmed to bind to HLA-B*35:01 and HLA-A*01:01, respectively. Therefore we only used these two monomers for flow cytometric staining and not the other HLA monomers.

To make this more clear, we have now adapted the following sentence in the respective paragraph (Supplementary Material, page 4):

“To confirm HLA-binding and T-cell recognition of the two predicted epitope candidates “TPSGTWLTY” and “CTDDNALAY(Y)”, HLA-peptide tetramers were produced from HLA-A*01:01 or HLA-B*35:01 monomers (easYmer, ImmunAware) by incubation with peptide for two days at room temperature and subsequent tetramerization with streptavidin-PE (Becton-Dickinson), according to the manufacturer’s protocol.”

SARS-CoV-2 specific IgG and IgA were negative throughout the course of the disease. They suggested that they tested for IgG and IgA against the S1 domain of the SARS-CoV-2 spike protein. And they added that they were negative throughout the course of disease. They are given as below in suppl. table. They were very low (lowest level: 2.11 g/l (reference: 7-16 g/l)), not only below normal range. They possibly varied due to the convalescent plasma recurrently given to the patient.

They added in the manuscript that IgG and IgA against the S1 domain of the SARS-CoV-2 spike protein were negative throughout the course of disease. How can the authors wait it to be positive when their patients’ antibody levels are as low as an agammaglobulinemic (generally, 1.0-2.0 g/l) patient. In such a patient with low IgG, the evaluation of the specific antibodies does not make sense.

Response: We agree that it was not necessarily expected that spike-specific antibody tests would turn out positive, and we stated this in our text (“As expected...”). Nonetheless, this was an important test to perform, and it might have been considered negligent to not even test for the presence of SARS-CoV-2-specific antibodies, especially since the patient received multiple infusions of convalescent plasma. A measurable increase of SARS-CoV-2-specific antibodies in some recipients of convalescent plasma has been demonstrated (Herman et al., Nat. Commun. 2021, 12:6853; Duan et al., PNAS 2020, 117:9490). Thus, it seemed interesting and important to test this in our patient. In our view, there was no

harm done in performing these tests and mentioning their results in our paper. The patient had IgG levels between 2.1 and 4.2 g/l, which is not nothing, but corresponds to approximately 2/7 to 4/7 of the lower end of the healthy reference range. Thus, we are unable to understand the reviewer's assertion, if we understood it at all correctly, that it was nonsensical of us to test the patient for antigen-specific antibodies.

Although the authors, in the new title, ' SARS-CoV-2 evolution and escape from CD8 T cells and antibodies during chronic infection', better defined the mechanism of evolution and escape during the chronic infection in this immunodeficient patient, the emphasis on only CD8+ T cells could not be understood as we saw, in this patient, the results of the defect in protein antibody response, which is very much related with CD4+ T cell levels. So, the emphasis on HLA Class I alleles and CD8+ T cells in this manuscript did not make sense. The authors may better think on covering this important issue while they are revising their manuscript.

Response: At this point we don't agree with the view that "the emphasis on HLA class I alleles and CD8+ T cells in this manuscript did not make sense". For obvious reasons we are not able to provide a study with an entirely different focus during the second round of reviews of a paper. We do not very clearly understand what the reviewer means by the expression "results of the defect in protein antibody response, which is very much related with CD4+ T cell levels". The patient was clearly unable to raise antibodies due to a complete absence of B cells after anti-CD20 treatment. Thus, no SARS-CoV-2-specific antibodies could have been produced, whether or not there was an additional defect in functional antigen-specific CD4+ T cells. Thus, it seems highly likely that the presence or absence of CD4+ T cells in this patient was of less consequence than the presence or absence of CD8+ cytotoxic T cells. Moreover, our focus on CD8 T cells is justified by the following additional considerations:

- (1) The patient had more CD8+ T cells than CD4+ T cells.**
- (2) The CD8+ T cells were more activated.**
- (3) CD8+ but not CD4+ responses were earlier described to be associated with SARS-CoV-2 control in patients with hematological cancer and impaired B cell immunity (Bange et al., Nat Med 2021, 27:1280).**
- (4) CD8+ T cells play a major role in control of most virus infections and are thus of obvious interest.**
- (5) Escape from CD8+ T-cell recognition through mutation or epitope polymorphism is a well known mechanism in a variety of viruses such as HIV, EBV, HCV, LCMV and others.**
- (6) Our approach was therefore tailored to the analysis of CD8 T cells. However, if any evidence on immunoevasion of CD4 T cells would have emerged during our**

analyses, we would have pursued such findings too. However, no such evidence emerged.

Note: I could not see in the methods the details of the measurement of antibodies against the S1 domain of the SARS-CoV-2 spike protein. They may better add it to the methods to make things clearer.

Response: We apologise for not having provided more detailed information about the method we used. In the supplementary methods we have now added the details in a new paragraph on page 2:

“SARS-CoV-2 serology.

Patient sera samples were tested for IgG and IgA against the SARS-CoV-2 S1 subunit using a commercial ELISA-kit by Euroimmun (Lübeck, Germany) following the manufacturer's instructions. These assays were performed in the accredited routine diagnostics laboratories of the Max von Pettenkofer Institute, Munich, Germany.”

I thank the authors for adding the lymphocyte subset levels. It is clearly seen in the Table that there is severe lymphopenia, ranging from 74 and 452 per μl (1220-3560). In this situation, not only number of CD8+ T cells nor CD4+ T cells is sufficient to show an effect on SARS-CoV-2.

In previous evaluation, the authors possibly misunderstood my comment on lymphocyte subsets. The lymphocyte subset asked below was not the lymphocyte subset of the healthy convalescent plasma donor.

- Question: A regular protein antibody response is a T-cell-dependent response. The authors may add the lymphocyte numbers of the donor? Did they measure the lymphocyte subset levels of the patient?

Response: We have no information about the lymphocyte numbers of the respective healthy convalescent serum donors, but these donors had no underlying medical conditions, especially no immunodeficiency. We now present the patient's lymphocyte subset levels in a new Supplementary Table 2 and selected T-cell subsets in Supplementary Fig. 6.

We apologize for the misunderstanding on our behalf in our previous response. We hope that we have adequately answered the reviewer's request by providing the lymphocyte subsets of the patient.

Note: They wrote convalescent serum in the manuscript. Did they use serum or plasma for the patient?

Response: As therapeutic intervention and as stated throughout the manuscript, the patient in this study received convalescent plasma and not serum. However, we used serum samples from the convalescent donors for our neutralisation assays as correctly stated in the supplementary methods page 2.:

“Neutralising activity of serum samples taken from the donors of convalescent plasma on the day of donation were tested against an early pandemic isolate...”

The authors suggested that SARS-CoV-2 RNAemia itself has been shown to be a marker for severe COVID-19 associated with higher mortality. However the reason of SARS-CoV-2 RNAemia was possibly derived from the insufficient response of the immune system.

Response: We apologize for being unable to find such a statement in our manuscript.

They suggest that the data on the therapeutic efficacy of convalescent plasma are still controversial in particular in severe COVID-19. However, in patients with antibody deficiency its effect has been clearly demonstrated and the guidelines regarding SARS-CoV-2 management for patients with inborn errors of immunity (IEI) have already included this therapy. Given the presented patient has a severe hypogammaglobulinemia, she might have benefited if the mutant virus experienced by the donors was the same as the patient.

Response: We agree that there may be a beneficial role for administration of convalescent plasma with high neutralising titres at early stages of infection in selected immunocompromised patients (Writing Committee for the REMAP-CAP Investigators, JAMA. 2021 Nov 2;326(17):1690-1702, or Lang-Meli et al Journal of Clinical Immunology (2022) 42:253–265) but we are unaware of any general recommendation based on the negative results from RCTs of convalescence plasma so far.

There are other questions to be answered. The study evaluated a single immunosuppressed individual. Evolution of the SARS-CoV-2 in this study seems to be related to the incomplete cellular and humoral response of this specific patient. It is difficult to suggest that the virus may evolve in a such a fast way in an individual with a normal immune response involving specific antibody and specific T cells. This issue may be discussed in the discussion.

Response: We fully agree with this point. We have added a clarifying note to the end of our discussion: “...in particular in patients with chronic infection and incomplete immunity.”

Minor comments:

Suppl. Figure 1A: The 1a, 1b, 2 and 3 should be defined in the figure legend. Moreover, the day that the antiviral therapy, IVIG, convalescent plasma was received by the patient should be indicated.

Response: Thank you for the suggestion. We added these informations to the figure and figure legend.

Supplementary Figure 2 is very demonstrative to show the date of the mutations. Did the authors see any correlation with the therapies the patient received?

We would like to refer to Figure 1a for illustration of the mutations and the administration of convalescent plasma.

Supplementary Fig 4: The original and mutant epitopes are given from donor 3. However, I could not understand which one is original and which one is mutant from the table.

To make Suppl. Fig. 4 fully consistent with the other figures, we now indicate the mutant amino acid (T>I) in red.

Supplementary Figure 5: What did the authors expect when they use HLA-B*35:01 tetramer for the donor 1 and HLA-A*01:01 for donor 2? What was their conclusion?

Thank you for raising this issue, we are happy to clarify. We can see that our explanation in Results may have been too brief. We have therefore added some explanatory text to our explanation of Supplementary Fig. 5 in the results section. The experiment was performed to test whether mutant epitopes produced weaker tetramer staining intensity and/or a reduced number of tetramer-staining T cells, compared to wild-type epitopes. Each of these outcomes would have been in line with reduced T-cell recognition of viral epitopes and thus with evasion from CD8 T-cell recognition after mutation. A combination of both outcomes was in fact observed.

Page 3, Line 79: Chronic SARS-CoV-2 infection in a patient after ?immunosuppressive? cancer therapy.

The authors should not specify the cancer therapy as immunosuppressive. All chemotherapies leads to immunosuppression. Instead they could indicate the time passed after therapy.

We have specified this accordingly to “Chronic SARS-CoV-2 infection in a lymphoma patient on B-cell depleting immunochemotherapy.”

Reviewer #2 (Remarks to the Author):

The authors have adequately addressed all of my concerns and comments.

Reviewer #3 (Remarks to the Author):

The authors provided some additional analyses to address the comments. The major conclusions, that the virus mutates for antibody and CD8 T cell escape, in my opinion, is still not supported by the data presented. In particular, the conclusions that the virus evolves to evade CD8 t cell responses in the context of immunosuppression are not supported by the data. Notably, the patient has a much higher CD8 T cell response against peptide pools of viral epitopes compared to non-immunosuppressed convalescent donors, and the patients CD8 T cells also respond to several peptide epitopes in the mutant virus (Fig 2g) and there is a reduced response to one of the mutant epitopes. However, the CD8 t cell response overall is highly activated (Fig. 2b), and in vivo, the response is to multiple epitopes, so the conclusion that the virus is evading responses by CD8 T cells is incorrect. Moreover, there is scant analysis of the CD8 T cell response at other timepoints-- the data presented show only results from a late (day 145) timepoint which are quite robust. The authors would need to show that there is a net reduction of the CD8 T cell response over time to support their conclusion that the virus is evading T cell recognition. But they don't have these data, and there is no evidence that this phenomenon is occurring, so the bottom line of the study, including the title is not supported by the data. There may be multipel factors contributing to the persistence of virus in this complex patient, including that the CD8 t cell are functionally impaired, that the virus had infiltrated other tissues, and other complicating features of the underlying lymphoma and the fact that she had very few B cells from previous treatments.

Response: We endeavour to provide a point by point response to the concerns raised below.

“The authors provided some additional analyses to address the comments. The major conclusions, that the virus mutates for antibody and CD8 T cell escape, in my opinion, is still not supported by the data presented. In particular, the conclusions that the virus evolves to evade CD8 t cell responses in the context of immunosuppression are not supported by the data.

Response: The concept that CD8 T cells drive SARS-CoV-2 mutation is also supported by the recent study of Stanevich et al. (preprint <https://doi.org/10.21203/rs.3.rs-750741/v1>), who have investigated another case of SARS-CoV-2 pneumonia in a patient with B-cell malignancy and B-cell deficiency after therapy. In their patient, as in ours, dominant fixed virus mutations occurred in several sequences coding for CD8 epitopes, and they found that one of the epitopes thus affected was no longer recognized by CD8 T cells. The same sequence (PTDNYITTY) is also altered by mutation in our study, although we could not experimentally investigate this particular epitope. We have added a paragraph to our discussion where we discuss that study and its parallels to our work.

Notably, the patient has a much higher CD8 T cell response against peptide pools of viral epitopes compared to non-immunosuppressed convalescent donors,...

Response: We do not see how this would be an argument for or against CD8 T- cell immunoevasion in this patient. On balance, a high CD8 T- cell response to non-mutant

viral epitopes might make it more likely, rather than less likely, that the virus was under strong immune pressure, resulting in immunoevasive mutations in CD8 T-cell epitopes. Conversely, the absence of functional CD8 T- cell responses would make it less likely that the virus had experienced selective pressure by CD8 T cells. As we also explained in our revised version, while CD8 T cells specific for ancestral SARS-CoV-2 were rather highly represented among PBMCs in relative terms, this needs to be seen in the context of a lower absolute CD8 T- cell count in the patient than in healthy donors.

“...and the patients CD8 T cells also respond to several peptide epitopes in the mutant virus (Fig 2g)”

Response: Fig. 2g does not show that the patient's CD8 T cells "respond to several peptide epitopes in the mutant virus". Fig. 2g shows the proportion of CD8 T cells that bind one non-mutant epitope and two successive mutant variants of this epitope.

“...and there is a reduced response to one of the mutant epitopes.”

Response: Here we agree, and we add that this reduction is rather strong. This is an argument in favor of T-cell-driven mutation, not against it.

“However, the CD8 t cell response overall is highly activated (Fig. 2b),...”

Response: Again, we do not see how this is an argument against a functional role of the CD8 T cells. The inverse might be concluded.

“...and in vivo, the response is to multiple epitopes,...”

Response: This argument is not clear to us. We showed multiple mutant epitopes. So the presence of a response to multiple non-mutant epitopes does not argue against immune-mediated immune pressure.

“...so the conclusion that the virus is evading responses by CD8 T cells is incorrect...”

Response: The arguments above do not support this conclusion, since they are either neutral or favourable to the idea that the virus is evading responses by CD8 T cells, as explained above.

“Moreover, there is scant analysis of the CD8 T cell response at other timepoints-- the data presented show only results from a late (day 145) timepoint which are quite robust.”

Response: We agree that our analysis is limited. We regret that additional samples are not available for analysis. However, considering the shortage of material, we are of the opinion that the match between our hypothesis and the experimental results is surprisingly strong. For four of six initial mutations in non-S proteins, we demonstrated that they occurred in CD8 T-cell epitopes and weakened or abolished their recognition.

The likelihood of such an occurrence happening by chance might be hard to quantify but seems extremely low.

“The authors would need to show that there is a net reduction of the CD8 T cell response over time to support their conclusion that the virus is evading T cell recognition.”

Response: We disagree. It is not at all unlikely that CD8 T- cell responses may be maintained in the periphery for a number of months even if they are no longer recognizing the currently predominant virus strain. The patient had low total T- cell numbers compared to a healthy donor, so there was no expectation of a strong contraction of the immune response. Is it the expectation of the reviewer that CD8 T cells should rapidly disappear once their target antigen is gone? This is not a likely expectation, since even in COVID-19 convalescent persons with an otherwise full T- cell repertoire, studies have shown that the SARS-CoV-2-specific CD8 T-cell response does not significantly decline for several months after infection (Bilich et al., 2021; Cohen et al., 2021).

Moreover (and hypothetically), residual antigen from non-mutant virus might have persisted (e.g. in lymph nodes) for weeks or months after mutant strains became dominant, thereby contributing and this antigen might have contributed to maintenance of CD8 T- cell responses.

“But they don't have these data, and there is no evidence that this phenomenon is occurring, so the bottom line of the study, including the title is not supported by the data.”

Response: We don't have data showing a strong decline of CD8 responses during infection or after the mutations happened (we assume this is meant by "this phenomenon".) This is correct, but such a decline is not expected, for reasons explained above. Therefore, we stand by our interpretations.

“There may be multiple factors contributing to the persistence of virus in this complex patient,...”

Response: We agree. We showed multiple mutations across that both CD8 and antibody epitopes were mutated. Therefore, it is likely that both kinds of mutations contributed to persistence of the virus in addition to the comorbidities of the patient. This is a major conclusion of our study.

“...including that the CD8 t cell are functionally impaired,..”

Response: It is possible that some CD8 T cells were functionally impaired. But this would make it even harder to explain the presence of multiple mutations in CD8 T-cell epitopes. There was no evidence for generalized CD8 T- cell dysfunction, which might have led to problems with control of other viruses.

“...that the virus had infiltrated other tissues,...”

Response: This does not argue against CD8 T cell-mediated control. CD8 T cells are expected to contribute to control of a virus in various tissues.

“...and other complicating features of the underlying lymphoma...”

Response: ...which, nonetheless, did not lead to a loss of functional CD8 T- cell responses, as we demonstrate.

“and the fact that she had very few B cells from previous treatments.”

Response: ...which is an argument in favour of an enhanced role of CD8 T cells in control of infection, and therefore enhanced selective pressure favouring mutations in CD8 T-cell epitopes.

Overall, we do not state that the escape mutations against CD8 T-cell responses are the only explanation for the fatal outcome in this complex and severely ill patient. However, we are confident that the data presented in our study clearly supports the statement that viral evolution in this host was in part driven by CD8 T-cell responses.

REVIEWERS' COMMENTS

Reviewer #1 (Remarks to the Author):

The authors have revised their paper according to the suggestions.
Best regards

Reviewer #3 (Remarks to the Author):

The authors have specifically addressed and argued with my comments. I still do not agree with their interpretation and believe that the title and bottom line of their manuscript is conjecture and misleading.